# ON THE BENEFITS OF DEEP RL IN ACCELERATED MRI SAMPLING

## ABSTRACT

Deep learning approaches have shown great promise in accelerating magnetic resonance imaging (MRI) by reconstructing high quality images from highly under-sampled data. While previous sampling methods relied on heuristics, recent work has improved the state-of-the-art (SotA) with deep reinforcement learning (RL) sampling policies, which offer the possibility of long term planning and of adapting to the observations at test time. In this work, we perform a careful reproduction and comparison of SotA RL sampling methods. Surprisingly, we find that i) a greedily trained fixed policy can match or outperform deep RL methods and ii) find and resolve subtle variations in the preprocessing and reporting which previously made results incomparable across different works. Our results cast doubt on the added value of current RL approaches over fixed masks in MRI sampling and highlight the importance of leveraging strong fixed baselines, standardized reporting as well as isolating the source of improvement in a given work via ablations. We conclude with recommendations for the training and evaluation of deep reconstruction and sampling systems for adaptive MRI based on our findings.

## 1 INTRODUCTION

Magnetic resonance imaging (MRI) is a non-invasive, non-ionizing medical imaging method which has been widely adopted in clinical settings due to its with unmatched quality in soft tissue contrast. However, MRI suffers from long scanning times, which limits patient comfort, imaging quality as well as throughput (Zbontar et al., 2019). These issues have been a longstanding research direction in the community, and one prominent way to shorten scanning times is acquiring less measurements and use a reconstruction method to recover the full image, a setting known as *accelerated MRI*.

Spurred by the great gains in reconstruction quality (Muckley et al., 2020), recent works have moved away from using generic, non-learning based heuristics, such as variable-density sampling (VDS) (Lustig et al., 2007) to select the measurements to acquire. Instead, there has been a growing interest in strong, learning-based methods that tailor sampling policies to the reconstruction method and achieve even greater accelerations, and as a result, further speeding up imaging.

The state-of-the-art approaches rely mostly on the idea of pairing a reconstruction model with a patient-adaptive sampling model, where the former estimates a clean image from partial measurements, and the latter selects measurements that it predicts as likely to improve the reconstruction quality. The selection policy can be trained relying on techniques from reinforcement-learning (Jin et al., 2019; Pineda et al., 2020; Bakker et al., 2020), heuristics such as a model trying to estimate the Fourier space error of the locations to be acquired (Zhang et al., 2019) or end to end via the straight-through estimator (Van Gorp et al., 2021; Yin et al., 2021). Other approaches choose to directly parameterize a fixed mask and do not learn a policy via neural networks (Bahadir et al., 2019; Huijben et al., 2020b; Weiss et al., 2019).

In this work, we focus our investigation on the benefits of the policies trained using reinforcement learning, namely the contributions of Pineda et al. (2020) and Bakker et al. (2020). We reproduce their results and extend their evaluation with additional baselines and experimental settings, i.e. change of hyperparameters and preprocessing. Deep RL approaches have drawn attention because the policies can in principle offer two important benefits: i) long term planning and ii) adapting to the currently acquired image at test time. The work of Pineda et al. (2020) however seems to indicate that long term planning could be the most important component in deep RL, as their results show that a non-adaptive, long term planning policy model trained on the dataset can perform as well as an adaptive, long-term planning policy. On the contrary, the contribution of Bakker et al. (2020)

| Sampling Method | Learning-based | Adaptive | Long horizon |
|---|---|---|---|
| VDS (Lustig et al., 2007) | ✗ | ✗ | ✗ |
| LBCS (Gözcü et al., 2018) | ✓ | ✗ | ✗ |
| Jin et al. (2019) | ✓ | ✓ | ✓ |
| Bakker et al. (2020) | ✓ | ✓ | ✓/✗ |
| Pineda et al. (2020) | ✓ | ✓/✗ | ✓ |

Table 1: Methods that will be considered in the paper. Bakker et al. (2020) considered fixed vs. adaptive and greedy vs. non-greedy, Pineda et al. (2020) considered data specific vs. subject specific policies and also compared against greedy methods. VDS stands for Variable-Density Sampling (Lustig et al., 2007) a common heuristic which is not learning-based.

highlights the importance of adaptivity, as a greedy policy, that does not do long-term planning is found to closely match policies that do long term planning. This means that at time of writing there are two peer reviewed SotA papers in apparent conflict with another.

Our results synthesize this apparent conflict: *We observe that a simple, easy-to-train method that does not rely on deep RL, does not attempt long term planning and is by definition not adaptive can perform as well as the state-of-the-art approaches of Pineda et al. (2020); Bakker et al. (2020).* The fixed sampling policies are obtained by optimizing a fixed policy greedily optimization on a training set, following the Learning-based Compressed Sensing (LBCS) approach of (Gözcü et al., 2018; Sanchez et al., 2019).

This trend can be consistently observed on the fastMRI single-coil knee dataset (Zbontar et al., 2019), where we carry out evaluations across a variety of settings, whether full scale images, cropping the field of view, various mask designs used for training and various architectures. We observe that such small changes in the experimental pipeline can easily lead to *reversals* in the performance or in the conclusions, by having the RL approach matched or outperformed by LBCS. It is possible then to choose an experimental setting and performance metrics to support one's desired conclusion: that RL outperforms LBCS, that LBCS outperforms RL or that their difference is not significant.

Together with the observation that current SotA RL methods only add marginal value over the simple baselines at best, we hope that this work will highlight the urgent need for further discussions in the community about standardized metrics, strong baselines, and careful design of experimental pipelines to evaluate MRI sampling policies fairly.

To ensure reproducibility and facilitate the use of strong baselines in future work we will release our code upon acceptance of this paper.

## 2 BACKGROUND

This work focuses on the following inverse problem, where we seek to recover a signal $x \in \mathbb{C}^P$ from partial observations $y \in \mathbb{C}^N$, $N \ll P$ obtained by subsampling a unitary transform matrix $A \in \mathbb{C}^{P \times P}$:

$$y_\omega = P_\omega A x + \eta, \tag{1}$$

where $\eta \in \mathbb{C}^N$ is a noise vector, $\omega \subseteq [P] := \{1, \ldots, P\}$ is an index set of allowable sampling locations with cardinality $N$, $P_\omega$ is a diagonal matrix such that $(P_\omega)_{ii} = 1$ if $i \in \omega$, 0 otherwise. $P_\omega$ and $\omega$ are referred to as the *(sampling) mask*, as they control what locations are acquired in the original signal. This problem is inherently ill-posed, due to $N \ll P$, and our first goal will be to construct an estimate $\hat{x}_{\omega;\theta} = f_\theta(y_\omega, \omega)$ of the original signal $x$, where $f_\theta$ is a reconstruction method parameterized by $\theta$.

### 2.1 MRI FUNDAMENTALS

In MRI, observations are obtained in the Fourier space, also referred to as *k-space*. The acquisition of data happens sequentially, but the physical constraints of an MRI acquisition make it more efficient to observe entire columns or rows at once, a setting known as *Cartesian sampling* (Zbontar et al., 2019). Non-Cartesian sampling is also possible (see for instance Weiss et al. (2019); Lazarus et al. (2019) for recent references), but some sampling methods like e.g. the pixel level sampling in Bahadir et al. (2019); Yin et al. (2021) are not practical for 2D MRI, as they do not allow for a reduction of scanning time compared sampling full trajectories.

The acquisition of a column or row is known as a *readout*, and the complete procedure consists in acquiring $N$ readouts sequentially. A full acquisition would require $P$ readouts, and acquiring only $N$ of $P$ lines accelerates the scan by a factor $P/N$, adequately named *acceleration factor*, the inverse

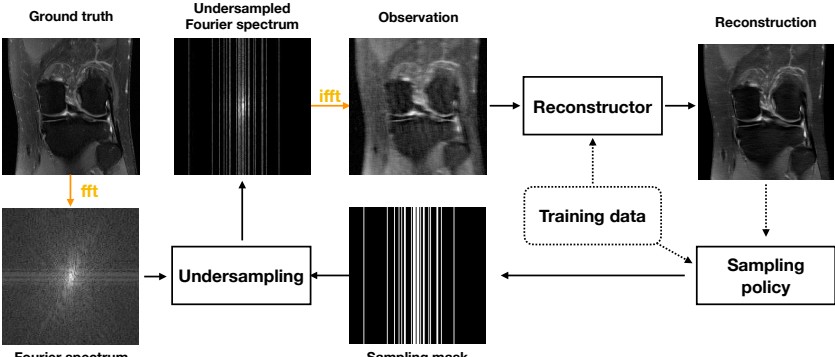

Figure 1: Overview of the accelerated MRI pipeline. Acquisition happens sequentially in Fourier space, where a policy decides on the next location to acquire. Dashed lines indicate optional relations: not all policies rely on training data, and not all policies are data adaptive, as shown in Table 1.

of which is known as the *sampling rate*, and the spectrum obtained with $N$ out of $P$ readouts is referred to as *undersampled*. The observations obtained by taking an inverse Fourier transform lead to an aliased image, and require processing through a reconstruction method.

In MRI, the Fourier space is typically structured, containing the bulk of energy in the low-frequencies located around the center of the space, and less around the high ones. While Compressed Sensing (CS) prescribes uniform sampling Donoho (2006); Candès et al. (2006), the structure of Fourier space made it necessary to leverage heuristics that assign more probability to low-frequencies in order to reflect the energy distribution. This approach is known as variable density-sampling (VDS) (Lustig et al., 2007).

### 2.2 SAMPLING OPTIMIZATION

The success of deep-learning approaches to MRI have led the broader medical imaging community to re-think the problem of optimizing the undersampling patterns in a data-driven fashion as well (Gözcü et al., 2018; Sanchez et al., 2019; Zhang et al., 2019; Jin et al., 2019; Pineda et al., 2020; Huijben et al., 2020a), instead of relying on heuristics such as variable density sampling (Lustig et al., 2007). The ideal sampling algorithm would tailor the mask to each instance of $\boldsymbol{x} \sim p(\mathbf{x})$ solving

$$\min_{\omega:|\omega| \leq N} \ell(\boldsymbol{x}, \hat{\boldsymbol{x}}_\theta(\boldsymbol{y}_\omega = \mathbf{P}_\omega \boldsymbol{A}\boldsymbol{x})), \tag{2}$$

which is not realizable since this requires using the unknown ground truth signal $\boldsymbol{x}$ at testing time and is computationally intractable due to the combinatorial nature of the problem. Two main approaches have been explored to circumvent this problem.

**Fixed (open-loop) sampling.** A majority of data-driven mask design approaches use fixed masks (Ravishankar & Bresler, 2011; Gözcü et al., 2018; Sanchez et al., 2019; Bahadir et al., 2019; Wu et al., 2019; Huijben et al., 2020b). The subsampling mask is constructed ahead of time - either via a heuristic or by using training data - and kept fixed at inference time. Formally, the problem of choosing the subsampling pattern corresponds to finding a subset $\omega$ that satisfies

$$\arg\min_{\omega:|\omega| \leq N} \mathbb{E}_{\boldsymbol{x} \sim p(\mathbf{x})} \left[ \ell(\boldsymbol{x}, \hat{\boldsymbol{x}}_\theta(\boldsymbol{y}_\omega = \boldsymbol{P}_\omega \boldsymbol{A}\boldsymbol{x})) \right], \tag{3}$$

where we are constrained with a maximal sampling budget $N$ and want to find a mask that minimizes a given loss function $\ell$. The true risk is substituted with the empirical one, estimated from training samples, and $\hat{\boldsymbol{x}}_\theta(\boldsymbol{y}_\omega = \boldsymbol{P}_\omega \boldsymbol{A}\boldsymbol{x})$ is an approximation of the ground truth obtained by reconstruction.

**Adaptive (closed-loop) sampling.** In contrast to Equation 3, adaptive sampling generates a dynamic sampling mask using a heuristic $\mathcal{H}_\phi$ evaluated at test time. For a fixed, unknown data sample $\boldsymbol{x}$, we use information of the previously obtained measurements $\boldsymbol{y}_{\omega_{t-1}}$ to determine which candidate $v \subset [P]$, $v \notin \omega_{t-1}$ should be acquired at time $t$. These methods solve at test time, for each individual target $\boldsymbol{x}$, the following sequence of problems:

$$v_t \in \arg\min_{v:v \in [P]} \mathcal{H}_\phi(v, \hat{\boldsymbol{x}}_{\omega_{t-1};\theta}, \omega_{t-1}) \quad \text{for } t = 1, \dots, N \tag{4}$$

where $\omega_t = \{\omega_{t-1}, v_t\}$ is defined in a nested fashion at each step. The heuristic score can be specifically trained for sampling leveraging reinforcement-learning framework (Bakker et al., 2020; Pineda et al., 2020; Jin et al., 2019), trained to directly estimate the current error and simply acquiring measurements where the error is estimated to be currently largest (Zhang et al., 2019), or be derived from available heuristics like posterior variance as in (Ji et al., 2008; Haupt et al., 2009; Sanchez et al., 2020).

## 3 THE MRI DATA PROCESSING PIPELINE

Section 2 outlined the mathematical setting underlying accelerated MRI, but does not capture practical considerations that can greatly affect the performance of methods as well as their applicability in the real world. In this section, we will step through the stages of the MRI data processing pipeline as shown in Figure 2. Note that this pipeline is independent of the sampling method and simply operates on a given mask.

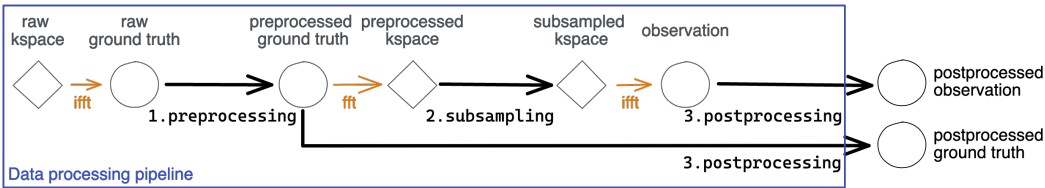

Figure 2: Illustration of the the data processing pipeline of MRI subsampling. Diamonds represent data in Fourier domain (k-space) and circles represent data in image domain. The postprocessed observation and ground truth are the data that are subsequently used for training the reconstruction and policy models. The pipeline features three main blocks, namely *preprocessing*, *subsampling* and *postprocessing*.

As we will show in Section 4.1, seemingly trivial changes at even a single of these stages can affect the results of the evaluation and lead to changes in the ordering of the performance of different policies, an observation that we will refer to as *reversal*. We mark sections where we observed changes leading to reversals with * and sections here they only shifted the results but do not lead to reversals with *italics*. We also discuss some caveats at these different steps in Appendix B

**Data sources and storage.** In all cases, the ground truth signal $x$ is initially acquired as a complex signal in k-space, generally using multiple coils. The fastMRI dataset (Zbontar et al., 2019) provides this raw multicoil data, as well as a simulated single-coil k-space which we use throughout this work.

**Preprocessing***. Due to computational constraints, it is common to resize the images by cropping and/or resizing or use magnitude images over the raw data. Cropping and resizing changes the ground truth distribution, which as seen in Table 3 can also lead to reversals in the final results.

**Sampling.** While in the real world the data is actually acquired by sampling k-space (prospective sampling), in practice, the acquisition is generally simulated by retrospectively undersampling fully sampled Cartesian data following Equation 1.

When training reconstructor and sampling method separately, training data are generally constructed with random masks that sample a certain fraction of center frequencies, and then the rest from a random distribution. There has been considerable variation in defining the parameters of these distributions, but to this day, no systematic study of their effect on the reconstruction quality have been carried out.

***Post processing and reconstruction.*** After sampling, the observation will be processed according to the implementation details of the reconstruction algorithm (e.g., normalized or standardized) and the reconstruction is computed. We discuss the impact of postprocessing in Appendices B and C.2.

**Evaluation metrics *.** Finally to judge the results, the most common metrics are peak signal-to-noise ratio (PSNR), mean squared error (MSE), normalized mean squared error (NMSE) and structural similarity index metric (SSIM).

All metrics tend to promote smooth images when used as a training loss (Muckley et al., 2020) but it is widely known that MSE, NMSE and PSNR focus on low-frequencies/high-energy components, which was also noted by Bakker et al. (2020). However, while MSE, NMSE and PSNR only operate on image differences and treats every pixel independently, SSIM is computed from local averages (Wang et al., 2004; Zbontar et al., 2019).

The metric is then reported as a curve plotted against sampling rate (Zhang et al., 2019), its reciprocal, the acceleration rate (Pineda et al., 2020) or aggregated by computing the average performance, the performance at end of sampling, or the area under curve (AUC). Details on the AUC computation are provided in Appendix A.4. As discussed throughout the experiments, the choice of metric and its aggregation can greatly impact the conclusions drawn from results.

## 4 RE-EXAMINING DEEP RL FOR MRI SAMPLING

Our initial impetus is the observation that the non-adaptive oracle used in Bakker et al. (2020) is highly reminiscent of the LBCS method (Gözcü et al., 2018), which has been used as a strong baseline in the literature (Jin et al., 2019; Sanchez et al., 2020). To our surprise, evaluating this methods against the greedy RL method in the data processing pipeline used in Zhang et al. (2019) resulted in the fixed method not only coming close to the non-adaptive oracle, but actually closely matching the RL method (see Table 3). Wanting to perform a fair comparison as well as to asses the impact of the variability in the pipelines used across the literature, we closely replicated the pipeline of Bakker et al. (2020)[1] as well as an extensive ablation study to understand this reversal and identify its source.

In the sequel, the term *setting* then refers to a particular choice of preprocessing, subsampling and postprocessing of the data, independently of the data source (we always use the same dataset) and the model or sampling policy used. Preprocessing and postprocessing are relevant throughout the training of the reconstruction and the policy models, whereas changes in the sampling masks only directly affect the pretraining of the reconstructor, as the training the policy is done by successive rollouts using the policy model itself.

**Dataset and preprocessing.** Like the original paper of Bakker et al. (2020), we used the fastMRI (Zbontar et al., 2019) single-coil knee dataset for the experiments. We slightly modify follow their preprocessing (using complex data instead of magnitude data, different data normalization) that do not affect the relative ordering of the different methods. We provide ablations and a detailed discussion of these changes in Appendix C.2.

Specifically, they crop the data to $128 \times 128$, which we refer to as (c)). Zhang et al. (2019) instead resizes the data to $128 \times 128$ which we replicate by first cropping to $256 \times 256$ and then resizing them to $128 \times 128$. This preprocessing results in images with different fields of view. We refer to this preprocessing as (c+r) and integrate it to our ablations. We also evaluate horizontal (h) sampling masks, used in Gözcü et al. (2018); Jin et al. (2019) in addition to the vertical (v) sampling masks used by Bakker et al. (2020); Zhang et al. (2019) and Pineda et al. (2020).

The deep reconstructors used in Pineda et al. (2020); Bakker et al. (2020) pre-trained by randomly sampling a mask from a set of distributions with different parameters. We ablated over two pretraining regimes which we abbreviate as b and z, respectively. We describe them in more detail in Appendix A

**Reconstruction models.** We ablate over the two reconstruction models used in the RL SotA (Pineda et al., 2020) and (Bakker et al., 2020), namely the cResNet architecture from Zhang et al. (2019), used in Pineda et al. (2020) and the U-Net baseline provided in the fastMRI dataset, used in Bakker et al. (2020). Hyperparameters and training details are discussed in Appendix A.3.

**Sampling methods** In each ablation setting we compared Bakker et al. (2020) to the following baselines and oracles (citations in squared brackets here refer to prior works that also evaluate them):

- Random sampling (**Random**): Acquire a fixed proportion of low-frequency lines in Fourier and then randomly sample the remaining lines [Jin et al., 2019; Pineda et al., 2020; Bakker et al., 2020].
- Low-to-high frequencies (**LtH**): select k-space lines from low-to-high frequencies lines [Zhang et al., 2019; Pineda et al., 2020; Jin et al., 2019].
- (Stochastic) Learning-based Compressive Sampling (**LBCS**) (Gözcü et al., 2018; Sanchez et al., 2019): This method trains a non-adaptive, greedy sampling policy that selects as a measurement candidate in each acquisition step the column that leads to the greatest average improvement over a sample from the training dataset. We use the stochastic version that scales better to large dataset and images [Jin et al., 2019].
- Non-adaptive Oracle (**NA Oracle**) (Bakker et al., 2020): This oracle is computed by training and evaluating LBCS directly on the test set, and illustrates the benefit of adaptivity in greedy methods. This is the instance of a non-adaptive, greedy sampling method [Bakker et al., 2020].

For the training of the policy models of Bakker et al. (2020), we use the parameters of the greedy model in their paper, which we refer to simply as **RL** in the sequel. We excluded the non-greedy version, as Bakker et al. (2020) notice that the performance of the non-greedy model with discount factor $\gamma = 0.9$ is always close to within one standard deviation of the greedy model, but significantly

---

[1]We thank the authors for providing the original checkpoints and general help and responsiveness during this replication

more computationally demanding. We studied both the short and long horizon sampling regimes but only report on the long horizon for conciseness, with the short horizon results summarized in Appendix C.1.

Except for the deterministic LtH and NA Oracle, we report the performance of each method averaged on three runs/separately trained RL policies, along with the standard deviation.

Summarizing, we start in the setting of Bakker et al. (2020) except for using complex data and not magnitude data and then ablate over: **i)** cropped (c) vs cropped+resized (c+r), **ii)** vertical (v) vs horizontal (h) Cartesian sampling, **iii)** reconstruction with a pretrained UNet (Ronneberger et al., 2015; Bakker et al., 2020) or a cResNet (Zhang et al., 2019) and **iv)** the training regime for the reconstructor proposed in Bakker et al. (2020) (b) or in Zhang et al. (2019) (z). The base setting is referred to as cvb, for cropped (preprocessing) + vertical (mask) + Bakker et al. (sampling parameters) using a both a UNet and a cResNet reconstructor. Our main results will compare the difference between cvb and c+rhz to illustrate the impact of largely different data processing. A more complete ablation, using the setting cvz to ablate over the impact of different mask parameters, c+rvz to study the impact of different field of views and mask orientation is carried out in Appendix C.3.

## 4.1 RESULTS ON BAKKER ET AL. (2020)

Comparing Table 2 and Table 3, that display respectively the performance at the end of the acquisition trajectory (25% sampling rate, as used in Bakker et al. (2020)) and the AUC over the whole trajectory, a *reversal* can clearly be seen on the c+rhz setting: on the first table, LBCS dominates in both the UNet and cResNet, but on the second, it matches RL for the UNet and is outperformed by RL for the cResNet. Conclusions drawn from the results are reversed or invalidated by using a different way to aggregate the results. Reversals also occur on both tables when comparing the cvb and c+rhz settings for each reconstruction method.

A consistent trend from Tables 2 and 3, as well as the ablations in Appendix C.3, is that the return on investment (ROI) of adopting RL over LBCS is generally marginal compared to what changes in the modeling pipeline can bring. Improving the reconstruction architecture or using masking regimes adapted to the sampling horizon yield much more significant gains, e.g., moving from a UNet to a cResNet brings an order of magnitude more improvement (around 0.01 SSIM difference) than what RL brings over LBCS (around 0.0015 at best).

We also observe that the performance of LBCS always remains close to the NA Oracle, testifying to the generalization ability of the fixed LBCS masks as indicated by theory. Other reversals can be observed in both cases when changing the data processing for both the SSIM at 25% and the AUC evaluations.

Finally, note that the large gap in performance between the cvb and the c+rhz settings is not due to a significant difference in performance of the reconstruction methods, but rather originates from the difference in field of view. This can be seen by comparing the SSIM value of the observations (before reconstruction) using the deterministic LtH policy in both cases: in the cvb setting, one gets an AUC of 0.4989 for the observation SSIM, whereas in the c+rhz setting, the AUC is 0.6534.

| Policy | cropped, vert., Bakker | | cropped+resized, horiz., Zhang | |
|---|---|---|---|---|
| | UNet | cResNet | UNet | cResNet |
| **Random** | $0.5249 \pm 0.0001$ | $0.5432 \pm 0.0004$ | $0.6567 \pm 0.0003$ | $0.6725 \pm 0.0006$ |
| **LtH** | 0.5832 | 0.6197 | 0.7325 | 0.7714 |
| **LBCS** | $0.6294 \pm 0.0009$ | $\mathbf{0.6417 \pm 0.0011}$ | $\mathbf{0.7768 \pm 0.0000}$ | $\mathbf{0.7941 \pm 0.0000}$ |
| **RL** | $\mathbf{0.6298 \pm 0.0002}$ | $0.6415 \pm 0.0002$ | $0.7761 \pm 0.0000$ | $0.7935 \pm 0.0001$ |
| **NA Oracle** | 0.6301 | 0.6428 | 0.7771 | 0.7942 |

Table 2: SSIM at 25% on the test set, on knee data comparing two models in the long horizon setting of Bakker et al. (2020), for the cvb (cropped, vertical, Bakker-type of masks) and c+rhz (cropped+resized, horizontal, Zhang-like masks) settings. This is an alternate aggregation as the one done on Table 3.

## 4.2 EXAMINING LONG RANGE ADAPTIVITY WITH PINEDA ET AL. (2020)

Given these results, we can observe that if there is indeed a benefit to adaptive but still greedy policies, it is highly sensitive to the parameters used, and the improvement over the fixed greedy baseline is

| Policy | cropped, vert., Bakker | | cropped+resized, horiz., Zhang | |
|---|---|---|---|---|
| | UNet | cResNet | UNet | cResNet |
| **Random** | $0.4348 \pm 0.0001$ | $0.4432 \pm 0.0002$ | $0.5860 \pm 0.0002$ | $0.5885 \pm 0.0002$ |
| **LtH** | 0.4849 | 0.5107 | 0.6678 | 0.6902 |
| **LBCS** | $0.5134 \pm 0.0004$ | $\mathbf{0.5243 \pm 0.0003}$ | $\mathbf{0.7035 \pm 0.0001}$ | $0.7096 \pm 0.0001$ |
| **RL** | $\mathbf{0.5142 \pm 0.0001}$ | $0.5242 \pm 0.0002$ | $\mathbf{0.7035 \pm 0.0002}$ | $\mathbf{0.7111 \pm 0.0009}$ |
| **NA Oracle** | 0.5140 | 0.5247 | 0.7038 | 0.7099 |

Table 3: AUC on the test set using SSIM, on knee data comparing two models in the long horizon setting of Bakker et al. (2020), for the `cvb` (cropped, vertical, Bakker-type of masks) and `c+rhz` (cropped+resized, horizontal, Zhang-like masks) settings. This is an alternate aggregation as the one done on Table 2.

marginal. One might wonder whether there will be a significant gain from adaptive RL policies if they are trained to perform long term planning on a longer horizon? To investigate this, we replicated the second SotA RL method described in Pineda et al. (2020), using the pretrained checkpoints due to computational constraints. This work uses a discount factor of $\gamma = 0.5$ and operates on the full fastMRI image of size $640 \times 368$, meaning it is both trained for a longer time horizon as well as having a lot more leeway for decision making. We replicated the *extreme* setting which starts with 2 center frequencies ($0.6\%$ sampling rate or $166\times$ acceleration) and is evaluated up to 100 frequencies ($30.1\%$ sampling rate or $3.32\times$ acceleration)[2]. As can be seen in Table 4, the LBCS mask is very close to the performance of both the data and sample specific DDQN policies provided by Pineda et al. (2020).

Interestingly, as seen in Figure 3 and Table 4, this is another example of a simple change in metric leading to a possible reversal of interpretation: visually in Figure 3a and when considering the AUC over sampling rate, LBCS seems competitive with the DDQN policies. In Figure 3b and the next column, we simply adopt the convention of Pineda et al. (2020) of using the acceleration rate (reciprocal of sampling rate) and we observe another reversal.

This puts more emphasis on the sub $5\%$ range (i.e. below 17 lines) where LBCS has not yet caught up on DDQN (see Figure 7 in Appendix F for a more expensive large-scale version of LBCS trained on MSE which *does* catch up). Finally, if we report only the final sampling rate as is the convention in Bakker et al. (2020), the presentation gives the win to LBCS again as can be seen in the rightmost column. We also performed a more detailed comparison of the masks given by the policies, using a subset of the first 200 test set images, keeping the order fixed in across methods. The results can be found in Appendix D.1 but can be summarized as **i)** the adaptivity of SS-DDQN mainly affects the ordering of frequencies, a large section of acquired frequencies being shared across samples and in similar regions to the LBCS mask at the final sampling rate **ii)** it confirms that the adaptive masks have a small edge only until about a sampling rate of $5\%$, after which LBCS catches up and overtakes the RL policy.

| Policy | SSIM | | | PSNR | | |
|---|---|---|---|---|---|---|
| | Samp. rate | Acc. factor | Final rate | Samp. rate | Acc. factor | Final rate |
| **Random** | 0.5801 | 0.4497 | 0.6723 | 26.489 | 22.327 | 28.962 |
| **LtH** | 0.5636 | 0.4506 | 0.6686 | 27.169 | 23.133 | 29.360 |
| **LBCS** | 0.6079 | 0.4787 | **0.6886** | **28.491** | 23.799 | **30.211** |
| **DS-DDQN** | 0.6101 | **0.4797** | 0.6855 | 28.240 | **23.978** | 29.652 |
| **SS-DDQN** | **0.6139** | **0.4797** | 0.6882 | 28.424 | 23.918 | 29.929 |
| **Adaptive Oracle** | 0.6341 | 0.4910 | 0.7131 | 29.013 | 24.498 | 30.683 |

Table 4: AUC on the test set when calculated against *sampling rate* and *acceleration* factor (1/sampling rate), as well as performance at the final sampling rate (100 lines acquired out of 332) on the knee dataset, using the processing of Pineda et al. (2020).

---

[2]We thank the authors for providing us with the original scores and general responsiveness and helpfulness during this replication.

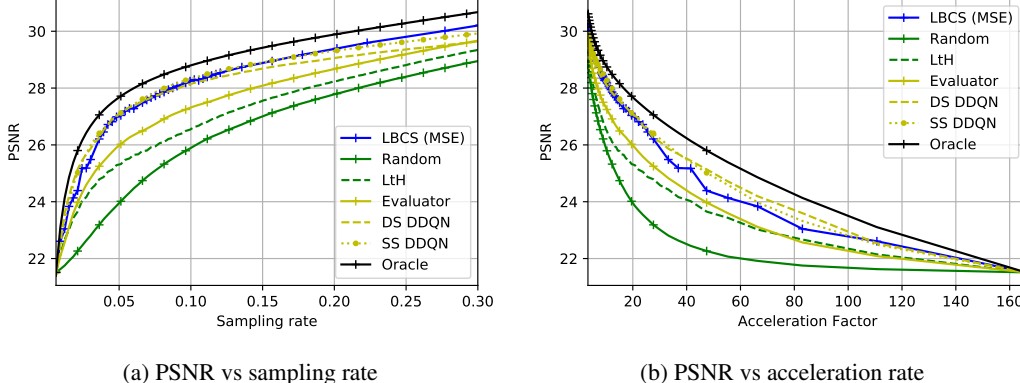

(a) PSNR vs sampling rate

(b) PSNR vs acceleration rate

Figure 3: PSNR performance plot on the test set, using the knee dataset with the processing from Pineda et al. (2020). The plots feature two ways to report the same result, which are also displayed in Table 4. The SSIM performance plot can be found in Figure 6 in the Appendix.

## 4.3 DISCUSSION

As we have seen in Section 4 and Section 4.2, the fixed mask of LBCS is competitive with or outperforms both the greedy policy gradient methods of Bakker et al. (2020) and the non-greedy DDQN methods of Pineda et al. (2020). Our ablations in Section 4 and Appendices C.3 and F, as well as the example of Figure 3 highlight that the benefit of current RL methods over the fixed baseline, if it exists at all, is so small that a change of field of view, architecture or mask distribution (cf. Tables 3 and 6) can lead to a reversal in the relative performance ordering.

Table 4 confirms the expectation that methods trained on SSIM can underperform with respect to PSNR and vice versa. In general agreement with the literature, we see that using several metrics is important to capture these tradeoffs, as reporting only one particular metric might hide underperformance in the other.

There have been various ways of reporting the metrics, ranging from showing sampling curves as in Figure 6a, reporting the area under curve (AUC) tables or simply reporting the metric at the final sampling rate. We see that reporting only the summarizing statistic can be deceitful and give relatively little information on the overall performance of the method. For instance, using only the last sampling rate, one could not distinguish between a method that performs well at all sampling rates leading to the final one, and one that has overall mediocre results and quickly improves on the performance at the end.

For this reason we consider sampling curves the gold standard for assessing the quality of accelerated MRI sampling methods, as a good performance at all sampling rates coupled with some stopping criterion could mean that MRI scans could be stopped when sufficient information has been acquired, further increasing their efficiency. If one wishes to report a single number, we recommend using AUC unless only the final performance is of interest.

There is also the question of reporting acceleration factors or sampling rates, which put focus on very different regimes of sampling: acceleration factors, as shown in (Pineda et al., 2020) focus on extreme undersampling rates. For instance, in the 2L scenario of Pineda et al. (2020), almost 90% of the plot and of the corresponding AUC consists of 20 lines out of the 100 acquired. While it is true that the high acceleration regimes are where it is most desirable to make improvements, results on sampling rates have the advantage of uniformly distributing the performance throughout a range of interest instead of a single acquisition determining a third of the metric as in fig. 3b.

Alternatives to using acceleration factors could consist in displaying sampling rates over a region of interest or computing the AUC by explicitly assigning more weight to low sampling rates instead of doing it implicitly and nonlinearly through acceleration factors.

Finally, a close study of the masks given by the SS-DDQN in Figure 5, the edge of the SS-DDQN over the LBCS mask in Figure 4a and the comparison of the LBCS and SSDQN edge over a fixed heuristic in Figures 4b and 4c reveals that the frequencies selected by the adaptive policy are very similar to those selected by the LBCS mask and that the variability between samples is concentrated at very early sampling rates.

### 4.4 RELATION TO OTHER WORKS

While we focused strictly on RL trained sampling policies leveraging pre-trained reconstructors, there are other paradigms for accelerated MRI sampling present in the literature.

The first line is that of using end-to-end training of masks using stochastic relaxations of the sampling mask to be able to differentiate it and optimize the reconstructor and sampling method jointly. Bahadir et al. (2019); Huijben et al. (2020) directly optimize for a single sampling rate and sampling mask. More recent work extend this technique for jointly training an adaptive policy, doing away with the pretraining and making more efficient use of the reconstructor's capacity (Yin et al., 2021; Van Gorp et al., 2021). Another line of joint training relies on self-supervised learning using Monte-Carlo Tree Search, as was done in Deepmind's AlphaGo Silver et al. (2017); Jin et al. (2019)

An open question regarding this line of research is whether the joint training enables to achieve a better sampling policy or simply serves as a curriculum and a way to specialize the sampling policy and reconstructor onto each other. To our knowledge, neither Van Gorp et al. (2021) nor Yin et al. (2021) investigated this. An interesting experiment could be to train a LBCS sampling mask on the co-trained reconstructor to see whether it can recover the sampling performance of the co-trained sampling network.

More similar to the spirit of our work, the recent study of Shimron et al. (2021) tackles subtle biases induced by the use of stored data and improper processing, Edupuganti et al. (2020) investigated specifically uncertainty methods for MRI reconstruction.

For works focusing on RL, Henderson et al. (2018) performed extensive ablation studies showing the sensitivity of RL methods to even minute variations in parameters. Engstrom et al. (2020); Gronauer et al. (2021) showed that the improvements of SotA RL methods can be traced to the exploitation of a subset of implementation details and algorithmic improvements and that even simple algorithms leveraging this subset can achieve SotA performance, similar to our work. This showcases that adapting to the data distribution seems to be sufficient to reach SotA in accelerated MRI sampling.

## 5 CONCLUSION

Taken together, our observations lead us to conclude the apparent conflict between the works of Bakker et al. (2020) and Pineda et al. (2020) is simply becasue at least *in their current state*, neither of the RL SotA methods do not offer significant benefits over fixed, greedily trained masks. Since greedy algorithms tend to perform near optimal in settings with submodularity (Krause & Golovin, 2014), we conjecture a similar structure might be present in problems like the MRI sampling problem. Determining whether this conjecture holds or whether the lack of added value originates from specific RL algorithm and their training remains to be determined and should be the focus for any researcher set on applying RL to such a problem.

Our results also enable us to provide practical advice, summarized below. We provide a more extensive discussion of these statements in Appendix E.

- Focus on improvements in the reconstructor architecture, mask distribution and algorithms used for training the reconstructor.
- Compare against strong baselines, such as LBCS.
- Show sampling curves and use AUC to aggregate your results instead of performance at the final sampling rate.
- Be mindful about preprocessing settings when evaluating a policy model. We recommend using the cropped+vertical setting with the data normalization implemented by Zbontar et al. (2019).

We also want to emphasize that conducting the experiments in this paper would have been impossible if the RL methods had not been exemplary in terms of openness and reproducibility. Without access to the checkpoints and code, and without the authors' responsiveness we would not have been able to reproduce both works and add the missing baseline. Despite the theoretical guarantees of LBCS, we were surprised that it matched and sometimes simply outperformed more sophisticated methods. We therefore do not view our work as criticism of these works but rather as an extension and a synthesis, and urge any future work to follow their lead in publishing codes, checkpoints and data.

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

# A IMPLEMENTATION DETAILS

## A.1 PRETRAINING REGIMES

The deep reconstructors used in Pineda et al. (2020); Bakker et al. (2020) are pre-trained by randomly sampling a mask from a set of distributions with different parameters. Bakker et al. (2020) uses a discrete set of sampling rates at $(25\%,25\%,25\%,\ 16.7\%,16.7\%,$ and $12.5\%)$ with $(25\%,16.7\%,12.5\%,$ $16.7\%,12.5\%,12.5\%)$ of the total selected frequencies being allocated to center frequencies and the rest sampled uniformly,while Zhang et al. (2019) uses a distribution where 10 center frequencies are always acquired, and between 0 and 38 additional frequency lines are acquired following a uniform distribution (as a reminder, the total number of possible sampling locations is 128 lines or columns).

## A.2 COMPUTATIONAL HARDWARE

We performed all of your experiments on a DGX-2 server using $A100$ GPUs. On this machine the reconstrction model of Pineda et al. (2020) fits into GPU memory with an effective batchsize of 50 (we use subbatching to enable arbitrary batch sizes) and it took $\approx 3$ hours to train each of the reported LBCS masks. Meanwhile masks for the comparisons of Bakker et al. (2020) could be trained in $\approx 20$ minutes, while training the RL policy took on the order of days as reported in the authors original paper. While one might be able to obtain a certain speedup using a more optimised or parallel RL algorithm, a key bottleneck is the sequential nature of the optimization.

## A.3 RECONSTRUCTOR ARCHITECTURES

We experiment on two main generator architectures, a UNet (Ronneberger et al., 2015) as used in Zbontar et al. (2019) and Bakker et al. (2020) and the cascade of residual networks, called cResNet introduced by Zhang et al. (2019).

For the cResNet in the comparison with Pineda et al. (2020) we used the checkpoint provided at https://facebookresearch.github.io/active-mri-acquisition/misc.html by the authors. All UNet models of the ablations of the Bakker et al. (2020) setting were trained following the hyperparameters described in the paper, i.e. using Adam Kingma & Ba (2014) and training to 50 epochs, otherwise using the hyperparameters from Zbontar et al. (2019) (except for the $10\times$ learning rate drop 40 epochs, we instead kept the initial $10^{-3}$ rate throughout). We used a batchs ize of 32.

For the cResNet models used in the ablations we used Adam as an optimizer with learning rate $10^{-3}$, betas of $(0.9, 999)$, trained for 50 epochs and a batch size of 32. The architecture is the same as in Zhang et al. (2019) except only outputting the mean of the image (no uncertainty channel), and using 72 channels for the 3 residual blocks, with $18, 36, 72$ channels in the encoder and $72, 36, 18$ channels in the decoder.

The images for both were undersampled using masks according to the settings described in appendix A.1. For the UNet, like Bakker et al. (2020) we take a single channel for magnitude only data as input, while for the cResNet we take the 2 complex channels as inputs. All models are trained using $\ell_1$ loss and directly output the reconstruction without a final nonlinearity. The UNet has $837'635$ parameters in total, while the c-ResNet is larger with $1'093'479$ parameters.

## A.4 AREA UNDER CURVE (AUC) COMPUTATION

We detail here the computation used to summarize performance curves with the AUC, used in Tables 3 and 4 in the body of the paper.

The area under curve is a numerical integration of the performance curve of the form $\{r_t, \ell(\boldsymbol{x}_i, \hat{\boldsymbol{x}}_\theta(\boldsymbol{y}_{\omega_t,i}))\}_{t=t_0}^T$ that relates the sampling rate or acceleration factor at the $t$-th step to its performance. We used the sklearn implementation, that implements it using a trapezoidal rule. We compute an individual AUC for each test sample and then aggregate the resulting set $\{\mathrm{AUC}_i\}_{i=1}^{n_\mathrm{test}}$ by computing its empirical mean and variance.

This measure is susceptible to changes in reparation of the area under the curve, and this is the reason that the result changes when moving from sampling rate to acceleration factor (1/sampling rate).

Indeed, when representing acceleration factors, in the case of Pineda et al. (2020) (cf. Table 4) 20 sampling locations cover $90\%$ of the acceleration factor plot, biasing the AUC towards attributing most of its weight the high acceleration factors.

## B    EXTENDED DISCUSSION ON THE MRI DATA PROCESSING PIPELINE

We extend the discussion of Section 4, focusing on some caveats and common pitfalls to avoid.

*Caveat #1 (Preprocessing).* Converting the raw ground truth to or starting from a magnitude induces a conjugate symmetry in Fourier space that is not present in real data, a distortion that makes our modeling less faithful to the physical model.

*Postprocessing details.* Typical normalization, as used for instance in Pineda et al. (2020), consist of dividing every input by a fixed value equal to the average energy of the dataset. Bakker et al. (2020) standardized their data and then clamped them to have standardized data in the range $[-6, 6]$.

We found that standardization used by Bakker et al. (2020) was necessary only when paired with non-residual models, which tracks with observations in the literature that residual networks can work without normalization, although usually this is studied for network internal normalization which we also leverage (see e.g. (Zhang et al., 2018) which removes normalization entirely by using a proper initialization).

*Caveat #2 (Postprocessing).* Normalization should occur in a consistent fashion between ground truth and observation data, as failing to do this can lead to inconsistent values in the reconstruction, especially in parts where data consistency occurs. In (Bakker et al., 2020), the authors normalize observations and ground truth using their respective statistics and denormalize the reconstructed image using the *ground truth* statistics, which is not compatible with the use of data consistency in the reconstructor. If one wishes to get a realistic estimate of the performance at deployment, it's also advisable to use only statistics available at test time (i.e. observation statistics, not ground truth statistics).

## C    FURTHER EXPERIMENTS ON BAKKER'S SETTING

### C.1    DATA PROCESSING DIFFERENCES

In section 4, we mentioned that we slightly deviated from the setting of Bakker et al. (2020). We now detail these changes and show that they do not change the *relative* performance of the different methods.

1. **Train-test split:** we used a different train-test split than Bakker et al. (2020), as we randomly splitted $10\%$ of the training set as a test set, and used the fastMRI validation set for test. We used the $50\%$ more central slices, resulting in 15599 training slices, 1743 validation slices and 3564 test slices.

2. **Complex data undersampling:** , Bakker et al. (2020) use magnitude ground truth images as the reference that is undersampled. We discussed in Appendix B the issue with this approach. We chose to use complex preprocessing of data, followed by taking magnitude of the observation obtained *after* undersampling.

3. **Data range:** PSNR and SSIM need to be provided with a maximal data range in their computation: Bakker et al. (2020) used the maximal intensity in the ground truth *volume*, while we used the maximal intensity of each ground truth *slice* or image.

4. **Data standardization:** Bakker et al. (2020) used a unique data standardization, where observations and ground truth data are standardized using their respective statistics, but denormalized, after reconstruction, using only the ground truth statistics. While this ensures a more closely matching data range, this introduces a mismatching data range that biases models leveraging data consistency, used in most state-of-the-art models. We performed *matched* data normalization and denormalization using the observation statistics for the observation, and the ground truth ones for the ground truth.

| Policy | Bakker's setting | Individual data range | Matched standardization | Mismatched setting | Complex preprocessing (matched - ours) |
|---|---|---|---|---|---|
| **Random (SH)** | *0.6251* | 0.5964 | 0.5798 | 0.5595 | 0.5608 |
| **LtH (SH)** | 0.6073 | 0.5762 | 0.5650 | 0.5617 | 0.5719 |
| **LBCS (SH)** | 0.6375 | 0.6099 | 0.5928 | 0.5684 | 0.5732 |
| **RL (SH)** | **0.6388** | **0.6112** | ***0.5941*** | **0.5698** | **0.5738** |
| **NA Oracle (SH)** | 0.6383 | 0.6107 | 0.5937 | 0.5717 | 0.5738 |

Table 5: AUC on the test set, using SSIM, on various data processing on Bakker's knee setting. The ablation is carried out over The mismatched setting refers to the evaluation of the model trained in the setting of Bakker et al. (2020), but evaluated complex data preprocessing instead of magnitude data preprocessing. This makes the reconstruction and policy models to be out of the distribution. A model trained on the complex preprocessing is reported in the complex preprocessing (matched - ours) column.

The third and fourth changes are related to postprocessing, and do *not* require retraining a model. We see on the three first columns of Table 5 that these changes do not alter the relative ordering of the methods. However, in absolute numbers, the worse performing method in the initial setting outperforms the best method the best performing one after these changes (highlighted in italic on Table 5). While comparing these numbers would be a mistake, this highlights the impact of subtle postprocessing changes and the need for care in comparing methods, especially across different papers.

## C.2 ABLATION OVER MAGNITUDE VS COMPLEX DATA

We first illustrate the impact of doing the appropriate preprocessing by evaluating the model of Bakker et al. (2020) using their data processing pipeline against a pipeline with complex preprocessing. The SSIM performance AUC is reported in Table 5.

This would correspond to a simulation of what would happen if the model of Bakker et al. (2020) was to be used in deployment, as one would receive undersampled complex observations that are only then transformed to magnitude images. While this does not induce any *reversal* in this case, we see that in the matched setting, the gap between LBCS and RL shrinks. We see also again that training and evaluating on the matched setting leads to larger performance improvements than the ones obtained by training sophisticated policies.

## C.3 FURTHER RESULTS IN THE BAKKER EXPERIMENT

We provide the results of our full ablation study on the components described in section 4. We report observations on the `cvb`, `cvz`, `c+rvz` and `c+rhz` settings. The summary of the full trajectory with an AUC is presented on Table 6 and the performance at the end of sampling is shown in Table 7.

It is difficult to consistently establish a trend for when reversals will happen, but several important observations can be made from the results.

First of all, by comparing Tables 6 and Table 7, we see that reversals can happen by considering one way of reporting over another (see `c+rvz` cResNet in both tables). This highlights again the impact of the way results are reported.

The effect of the masks used for pretraining the reconstructor is also interesting. Comparing the `cvb` and `cvz` results, once can consistently see the following trend. In the short horizon setting (LHS of the tables), using `cvb` leads to consistent improvements over `cvz`. In the long horizon setting, the opposite is true. Recall that the `b` masks are discretely distributed from sampling rates 12.5% to 25%, which matches the short horizon experiment range. The `z` masks span a continuous range, from roughly 7% to 37.5%, whereas the long horizon experiment spans a range from 3% to 25% sampling rate. These results suggest that matching the pretraining regime to the regime on which evaluation will be carried out has a significant influence on the performance of the sampling policy, an observation which has, to our knowledge, never been quantified before.

**PSNR evaluation.** We also provide a PSNR evaluation for the models trained on SSIM in Tables 8 and 9. There does not seem a clear trend or correlation between the policy's performance on SSIM

| Policy | Short Horizon | | | | Long Horizon | | | |
|---|---|---|---|---|---|---|---|---|
| | cvb | | c+rhz | | cvb | | c+rhz | |
| | UNet | CResNet | UNet | CResNet | UNet | CResNet | UNet | CResNet |
| **Random** | 0.5608 | 0.5659 | 0.7292 | 0.7338 | 0.4348 | 0.4432 | 0.5860 | 0.5885 |
| **LtH** | 0.5719 | 0.5764 | 0.7309 | 0.7412 | 0.4849 | 0.5107 | 0.6678 | 0.6902 |
| **LBCS** | 0.5732 | 0.5828 | **0.7430** | **0.7526** | 0.5134 | **0.5243** | **0.7035** | 0.7096 |
| **RL** | **0.5738** | **0.5830** | **0.7430** | 0.7524 | **0.5142** | 0.5242 | **0.7035** | **0.7111** |
| **NA Oracle** | 0.5738 | 0.5832 | 0.7430 | 0.7527 | 0.5140 | 0.5247 | 0.7038 | 0.7099 |

| Policy | Short Horizon | | | | Long Horizon | | | |
|---|---|---|---|---|---|---|---|---|
| | cvz | | c+rvz | | cvz | | c+rvz | |
| | UNet | CResNet | UNet | CResNet | UNet | CResNet | UNet | CResNet |
| **Random** | 0.5580 | 0.5661 | 0.6851 | 0.6984 | 0.4483 | 0.4621 | 0.5252 | 0.5276 |
| **LtH** | 0.5663 | 0.5759 | 0.6739 | 0.7028 | 0.5122 | 0.5218 | 0.6292 | 0.6444 |
| **LBCS** | 0.5712 | 0.5822 | 0.7034 | 0.7235 | 0.5174 | 0.5328 | 0.6568 | 0.6712 |
| **RL** | **0.5717** | **0.5829** | **0.7042** | **0.7239** | **0.5183** | **0.5334** | **0.6582** | **0.6733** |
| **NA Oracle** | 0.5718 | 0.5835 | 0.7038 | 0.7236 | 0.5180 | 0.5336 | 0.6568 | 0.6717 |

Table 6: AUC on the test set, using SSIM, for the full ablation study using the model of Bakker et al. (2020). The short and long horizon results are *not* comparable with each other, as AUCs are integrated on the whole range of sampling rates. The top right part of the table (long horizon) replicated the results of table 3, excluding the standard deviation for legibility. The rest of the ablation were *not* averaged on several seeds for computational reasons. Recall that cvb stands for cropped, vertical lines, Bakker-like mask distribution, c+rhz stands for cropped then resized, horizontal lines and Zhang-like masks, cvz stands for cropped, vertical lines and Zhang-like masks and c+rvz stands for cropped then resized, vertical lines and Zhang-like masks.

| Policy | Short Horizon | | | | Long Horizon | | | |
|---|---|---|---|---|---|---|---|---|
| | cvb | | c+rhz | | cvb | | c+rhz | |
| | UNet | CResNet | UNet | CResNet | UNet | CResNet | UNet | CResNet |
| **Random** | 0.607 | 0.6141 | 0.7565 | 0.7667 | 0.5249 | 0.5432 | 0.6567 | 0.6567 |
| **LtH** | 0.6267 | 0.6313 | 0.7602 | 0.7786 | 0.5832 | 0.6197 | 0.7325 | 0.7714 |
| **LBCS** | 0.6288 | 0.6413 | **0.7751** | **0.7923** | 0.6294 | **0.6417** | **0.7768** | **0.7941** |
| **RL** | **0.6298** | **0.6417** | 0.7745 | 0.7921 | **0.6298** | 0.6415 | 0.7761 | 0.7935 |
| **NA Oracle** | 0.6301 | 0.6421 | 0.7751 | 0.7926 | 0.6301 | 0.6428 | 0.7771 | 0.7942 |

| Policy | Short Horizon | | | | Long Horizon | | | |
|---|---|---|---|---|---|---|---|---|
| | cvz | | c+rvz | | cvz | | c+rvz | |
| | UNet | CResNet | UNet | CResNet | UNet | CResNet | UNet | CResNet |
| **Random** | 0.6053 | 0.6148 | 0.7201 | 0.7401 | 0.5502 | 0.5693 | 0.6191 | 0.6359 |
| **LtH** | 0.619 | 0.6309 | 0.7052 | 0.7487 | 0.6190 | 0.6309 | 0.7052 | 0.7487 |
| **LBCS** | 0.6282 | 0.6413 | 0.7411 | 0.7710 | 0.6293 | **0.6451** | 0.7453 | **0.7746** |
| **RL** | **0.6288** | **0.6416** | **0.7417** | **0.7714** | **0.6304** | 0.6452 | **0.7470** | 0.7738 |
| **NA Oracle** | 0.6292 | 0.643 | 0.741 | 0.7717 | 0.6304 | 0.6464 | 0.7457 | 0.7755 |

Table 7: SSIM at 25% sampling rate, using SSIM, with the model of Bakker et al. (2020). This is the counterpart of the results shown in Table 6, where the acronyms used are explicited. Here, the results across short and long horizon are comparable.

and PSNR. We observe however the same kind of dynamics in the settings, where cropped + resized data naturally have a higher PSNR than cropped ones, and cResNet improves the reconstruction quality of UNet.

| Policy | Short Horizon | | | | Long Horizon | | | |
|---|---|---|---|---|---|---|---|---|
| | cvb | | c+rhz | | cvb | | c+rhz | |
| | UNet | CResNet | UNet | CResNet | UNet | CResNet | UNet | CResNet |
| **Random** | 24.206 | 24.180 | 27.641 | 28.365 | 20.802 | 20.836 | 22.375 | 22.371 |
| **LtH** | 24.486 | 24.420 | 27.628 | 28.640 | 23.570 | 23.578 | 26.607 | 27.317 |
| **LBCS** | **24.493** | **24.517** | **28.254** | **29.154** | 23.574 | 23.635 | **27.220** | **27.802** |
| **RL** | 24.466 | 24.506 | 28.247 | 29.147 | **23.585** | **23.646** | 27.210 | 27.757 |
| **NA Oracle** | 24.510 | 24.521 | 28.257 | 29.152 | 23.591 | 23.639 | 27.222 | 27.757 |

| Policy | Short Horizon | | | | Long Horizon | | | |
|---|---|---|---|---|---|---|---|---|
| | cvz | | c+rvz | | cvz | | c+rvz | |
| | UNet | CResNet | UNet | CResNet | UNet | CResNet | UNet | CResNet |
| **Random** | 23.978 | 24.184 | 25.845 | 26.949 | 20.967 | 21.437 | 20.428 | 20.625 |
| **LtH** | 24.108 | 24.396 | 25.641 | 27.048 | 23.441 | 23.703 | 24.487 | 25.510 |
| **LBCS** | **24.269** | 24.490 | 26.515 | **27.870** | **23.600** | 23.846 | **25.307** | **26.316** |
| **RL** | 24.226 | **24.498** | **26.530** | 27.861 | 23.581 | **23.858** | 25.240 | 26.246 |
| **NA Oracle** | 24.271 | 24.513 | 26.528 | 27.856 | 23.603 | 23.848 | 25.309 | 26.250 |

Table 8: AUC on the test set, using PSNR, for the full ablation study using the model of Bakker et al. (2020). The short and long horizon results are *not* comparable with each other, as AUCs are integrated on the whole range of sampling rates. The rest ablation were *not* averaged on several seeds for computational reasons. Recall that cvb stands for cropped, vertical lines, Bakker-like mask distribution, c+rhz stands for cropped then resized, horizontal lines and Zhang-like masks, cvz stands for cropped, vertical lines and Zhang-like masks and c+rvz stands for cropped then resized, vertical lines and Zhang-like masks.

| Policy | Short Horizon | | | | Long Horizon | | | |
|---|---|---|---|---|---|---|---|---|
| | cvb | | c+rhz | | cvb | | c+rhz | |
| | UNet | CResNet | UNet | CResNet | UNet | CResNet | UNet | CResNet |
| **Random** | 24.583 | 24.604 | 28.241 | 29.113 | 21.520 | 21.684 | 23.547 | 23.843 |
| **LtH** | 25.056 | 24.984 | 28.176 | 29.567 | 25.056 | 24.984 | 28.176 | 29.567 |
| **LBCS** | **25.068** | **25.117** | **29.088** | 30.214 | 25.043 | 25.077 | **29.226** | **30.234** |
| **RL** | 25.032 | 25.108 | 29.050 | **30.218** | **25.071** | **25.101** | 29.194 | 30.188 |
| **NA Oracle** | 25.095 | 25.120 | 29.090 | 30.233 | 25.111 | 25.111 | 29.211 | 30.229 |

| Policy | Short Horizon | | | | Long Horizon | | | |
|---|---|---|---|---|---|---|---|---|
| | cvz | | c+rvz | | cvz | | c+rvz | |
| | UNet | CResNet | UNet | CResNet | UNet | CResNet | UNet | CResNet |
| **Random** | 24.369 | 24.623 | 26.657 | 27.957 | 21.791 | 22.497 | 21.854 | 22.506 |
| **LtH** | 24.551 | 24.961 | 26.223 | 28.143 | 24.551 | 24.961 | 26.223 | 28.143 |
| **LBCS** | **24.817** | 25.089 | 27.439 | **29.069** | 24.850 | **25.084** | 27.518 | **29.145** |
| **RL** | 24.765 | **25.102** | **27.451** | 29.069 | **25.102** | 24.807 | **27.583** | 29.041 |
| **NA Oracle** | 24.813 | 25.123 | 27.429 | 29.043 | 24.837 | 25.101 | 27.446 | 29.072 |

Table 9: PSNR at 25% sampling rate, using PSNR, with the model of Bakker et al. (2020). This is the counterpart of the results shown in Table 8, where the acronyms used are explicited. Here, the results across short and long horizon are comparable.

# D DIGGING INTO LONG RANGE ADAPTIVITY

Since we had to update the codebase to the current Pytorch FFT API we used the Low-To-High baseline as confirmation that we indeed replicate their setting exactly and then compare their method against the Low-To-High and random sampling baselines. We also use the fact that the code provided by the authors reports the images used for evaluation to evaluate LBCS on *exactly* the same subset of the validation set as the original codebase (we always train on the training set only). Similar Pineda et al. (2020), we train a separate LBCS mask for the reporting of SSIM and PSNR/MSE/NMSE (we found the latter 3 to be highly correlated in performance).

## D.1 ADAPTIVITY

As can be seen in fig. 5a., d. and e, LBCS puts more emphasis on the center frequencies, but acquires similar sections of k-space as the SSDDQN. It also creates a more symmetric masks, which is in line with Pineda et al. (2020) observations that SSIM creates more asymmetric masks. More interestingly, as can be seen in fig. 5c., most of the variation is concentrated at the early sampling rates (left of the plot) with the std and especially the coefficient of variation ($\frac{\sigma}{\mu}$) decaying towards zero in most locations. This implies that while SSDDQN is indeed adapting to each image individually, this mainly affects ordering early on and after 20 -40 samples ($6 - 12\%$ sampling rate) LBCS starts to catch up. This is also supported by two observations:

1. LBCS underperforms SSDQN by a larger margin than on the full evaluation, which we interpret as the mask being good *on average* and with a smaller sample there is a higher chance of individual suboptimality

2. the per-image difference between LBCS and SSDQN grows in favor of SSDDQN until about 40 samples where LBCS starts to slowly recover. The distribution of the difference however becomes much wider, implying there are images where LBCS performs wildly different from SSDDQN

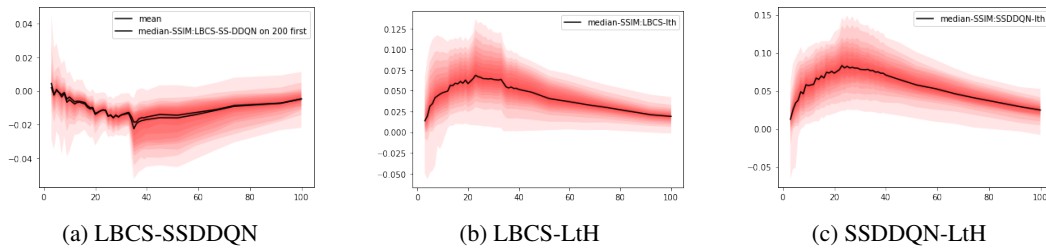

|       (a) LBCS-SSDDQN       |       (b) LBCS-LtH       |       (c) SSDDQN-LtH       |

Figure 4: Per Image distribution of SSIM differences between sampling methods across the sampling process. Each shade corresponds to a 10 percent region, with the lightest shade indicating max and min regions.

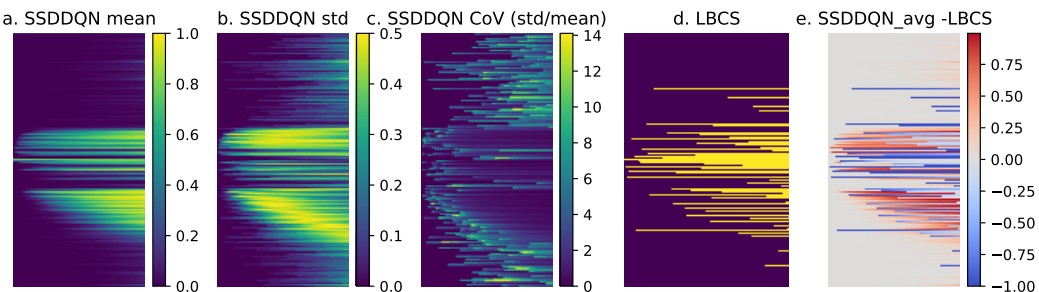

Figure 5: SS-DDQN variability and comparison with the LBCS mask. CoV stands for Coefficient of Variation here.

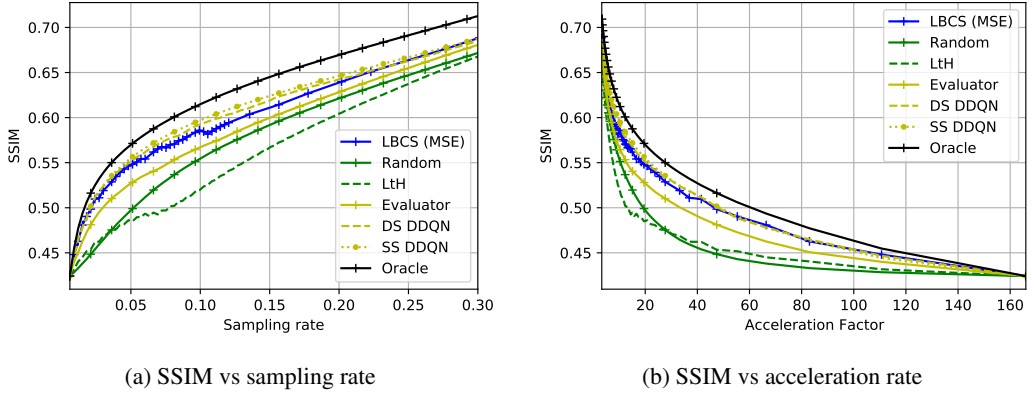

(a) SSIM vs sampling rate

(b) SSIM vs acceleration rate

Figure 6: Two ways to report the same result, SSIM version of fig. 3

# E    DETAIL ON THE PRACTICAL RECOMMENDATIONS

In this appendix, we discuss the recommendations issued in the conclusion, and provide the supporting evidence from our results.

## E.1    FOCUS ON IMPROVEMENTS IN THE RECONSTRUCTOR ARCHITECTURE, MASK DISTRIBUTION AND ALGORITHMS USED FOR TRAINING THE RECONSTRUCTOR

Our results and ablations on the setting of (Bakker et al., 2020) consistently show that the improvements obtained by changing the reconstructor architecture or the mask distribution are orders of magnitude more impactful than moving from LBCS to RL. This is supported the results of both Section 4.1 and Appendix C.3, where we see that moving from a UNet to a cResNet typically brings a significantly larger improvement (typically 0.01 SSIM) than what RL brings over LBCS in the best case (at most 0.0015 SSIM).This trend is also seen in Appendix C.3 when moving from `cvb` in the short horizon setting to `cvz` in the long horizon setting. We refer the reader to this section for more details.

## E.2    COMPARE AGAINST STRONG BASELINES, SUCH AS LBCS

This point is established throughout our paper, where all results illustrate that, at best, RL methods bring moderate improvement over LBCS. This improvement often comes at the cost of prohibitive computational expense, even on higher end DGX-2 servers[3], while training LBCS required at most a couple of hours (cf. Appendix A.2).

## E.3    SHOW SAMPLING CURVES AND USE AUC TO AGGREGATE YOUR RESULTS INSTEAD OF PERFORMANCE AT THE FINAL SAMPLING RATE

There is no consensus on how results should be aggregated from sampling curves. Bakker et al. (2020); Van Gorp et al. (2021) reported performance at the end of the acquisition, and Pineda et al. (2020) reported AUC curves computed on the acceleration factor. Gözcü et al. (2018); Yin et al. (2021) reported performance at selected sampling rates, and for other works such as Jin et al. (2019), it is not clear how results were aggregated.

We believe that reporting the AUC on sampling rates, computed on the whole range of acquisition steps allows to most meaningfully quantify the performance of a policy on its whole trajectory. It does not require to select a sampling rate at which the result should be evaluated, and using sampling rates as opposed to acceleration factor allows to equally weight the contribution of each acquired line.

Table 4 and Figure 3 compellingly illustrate that reporting the policy at a given sampling rate is not representative of its performance throughout the acquisition procedure.

## E.4    BE MINDFUL ABOUT PREPROCESSING SETTINGS WHEN EVALUATING A POLICY MODEL. WE RECOMMEND USING THE CROPPED+VERTICAL SETTING WITH THE DATA NORMALIZATION IMPLEMENTED BY ZBONTAR ET AL. (2019)

We discussed in Appendix B that Bakker et al. (2020) used a data normalization that is, among other things, incompatible with data consistency, a commonly used building block for cascading networks (Schlemper et al., 2018; Zhang et al., 2019). This can be prevented by using a normalization based on observation rather than ground truth statistics, as implemented in Zbontar et al. (2019).

Regarding the experimental setting, vertical masks have ubiquitously used on the fastMRI dataset Zbontar et al. (2019); Huijben et al. (2020); Bakker et al. (2020); Pineda et al. (2020) and cropping has been the most common preprocessing to alleviate the computational burden of the large images ($640 \times 368$) used in the dataset (Bakker et al., 2020; Huijben et al., 2020b; Yin et al., 2021). Evaluating models on cropped data with vertical masks will then facilitate reproducibility among different works. We would additionally recommend to researchers to evaluate their models on the cropped+resized in addition to the cropped only setting, as the images display a significantly different field of view (cf. Figures 8 and 10).

---

[3]The authors of (Pineda et al., 2020) confirmed to us that it took more than 20 days to train their model.

## F    DETAILS ON LBCS USED FOR THE FIXED MASK DESIGN

Algorithm 1 shows the pseudocode of the algorithm we used for designing the fixed masks used for the comparisons throughout the paper, while Table 11 summarizes the hyper parameters used and Table 10 gives an overview of the computational complexity. Note that LBCS is almost completely parallelizable, which leads to the stark runtime differences noted in Appendix A. As can be seen when comparing Figure 3 and Figure 7, once a certain data batch size $l$ and sampling candidate set cardinality $k$ is reached, LBCS performance saturates although it continues to benefit especially in the low sampling regime. This means that one could very feasibly reduce $l$ for the comparison with Bakker et al. (2020), we simply went with a larger batchsize because we could.

---

**Algorithm 1** Stochastic LBCS mask design
**(G)** refers to the greedy algorithm Gözcü et al. (2018)
**(SG)** refers to the stochastic greedy algorithm of Sanchez et al. (2019)

---

**Input**: Training data $\{\boldsymbol{x}\}_{i=1}^m$, reconstructor $f_\theta$, sampling set $\mathcal{S}$, max. cardinality $N$, samp. batch size $k$, train. batch size $l$, performance metric $\eta(\cdot, \cdot)$
**Output**: Sampling pattern $\omega$

1:
2: **while** $|\omega| \leq N$ **do**
3:     **(G)** $\begin{cases} \text{Pick } \mathcal{S}_{iter} = \mathcal{S} \\ \text{Pick } \mathcal{L} = \{1, \ldots, m\} \end{cases}$
4:     **(SG)** $\begin{cases} \text{Pick } \mathcal{S}_{iter} \subseteq \mathcal{S} \text{ at random, with } |\mathcal{S}_{iter}| = k \\ \text{Pick } \mathcal{L} \subseteq \{1, \ldots, m\}, \text{ with } |\mathcal{L}| = l \end{cases}$
5:     **for** $S \in \mathcal{S}_{iter}$ such that $|\omega \cup S| \leq \Gamma$ **do**
6:         $\omega' = \omega \cup S$
7:         For each $\ell \in \mathcal{L}$ set $\hat{\boldsymbol{x}}_\ell \leftarrow f_\theta(\boldsymbol{P}_{\omega'}\boldsymbol{A}\boldsymbol{x}_\ell, \omega')$
8:         $\eta(\omega') \leftarrow \frac{1}{|\mathcal{L}|} \sum_{\ell \in \mathcal{L}} \eta(x_\ell, \hat{x}_\ell)$
9:     $\omega \leftarrow \omega \cup S^*, \text{ where } S^* = \underset{S:|\omega \cup S| \leq N}{\arg\max} \ \eta(\omega \cup S)$
10: **return** $\omega$

---

| Method | Forward | Backward | Total |
|---|---|---|---|
| (Bakker et al., 2020) | $q(8)E(50)B(16)H(n_r + n_p) = 6'400H(n_r + n_p)$ | $6400Hn_p$ | $6'400H(n_r + 2n_p)$ |
| LBCS for Bakker et al. (2020) | $n(H)k(128)l(256)n_r = 32'768Hn_r$ | 0 | $32'768Hn_r$ |
| (Pineda et al., 2020) | $5e6(n_r + n_p)$ | $5e6(n_p)$ | $5e6(n_r + 2n_p)$ |
| LBCS for Pineda et al. (2020) | $n(100)k(64)l(20)n_r = 128e3n_r$ | 0 | $128e3n_r$ |

Table 10: Approximate computational cost of the compared methods. Note that at test time, LBCS is basically free while the RL policies will still need to be deployed. $n_r, n_p$ are the parameter counts of the reconstruction and sampling policies respectively

| **Setting** | Num. lines $|\mathcal{S}|$ | Max. cardinality $N$ | Candidate set size $k$ | Data batch size $l$ |
|---|---|---|---|---|
| Short horizon (Bakker et al., 2020) | 128 | 16 | 128 | 256 |
| Long horizon (Bakker et al., 2020) | 128 | 28 | 128 | 256 |
| (Pineda et al., 2020) | 332 | 100 | 64 | 20 |
| (Pineda et al., 2020) "big" | 332 | 48 | 200 | 256 |

Table 11: The hyperparameters used for the stochastic LBCS masks throughout comparisons

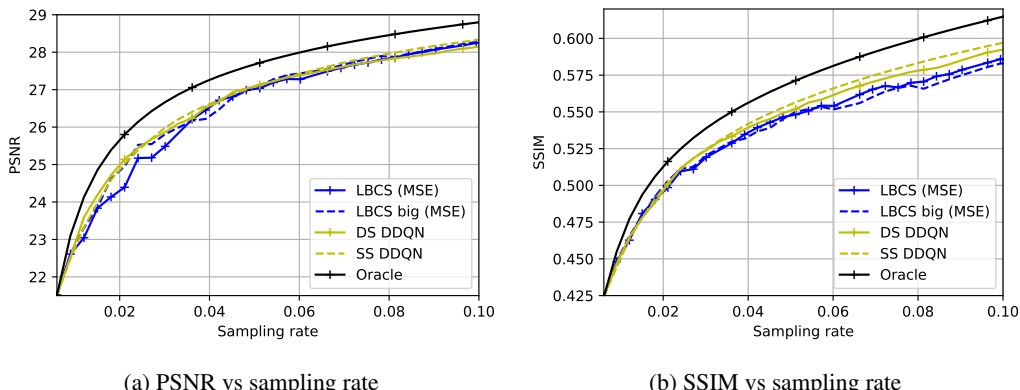

(a) PSNR vs sampling rate

(b) SSIM vs sampling rate

Figure 7: PSNR and SSIM of LBCS with larger batch size (LBCS - big) trained on MSE, zoomed to the region where the smaller batch size LBCS underperformed. Note that it fully matches or outperforms both versions of DDQN on PSNR.

# G  VISUAL COMPARISONS

In this section, we present a visual comparison over a selected set of models, policies, settings and images. We present a visual evaluation of the models and policies at sampling rates 25% and 12.5% on Figures 8 and 9. In addition, we present a more exhaustive set of reconstruction, at various sampling rates, on Figure 10, where we display both types of sequences that were used to generate the data, namely proton density (PD) and proton density, fat saturated (PDFS) of images (Zbontar et al., 2019).

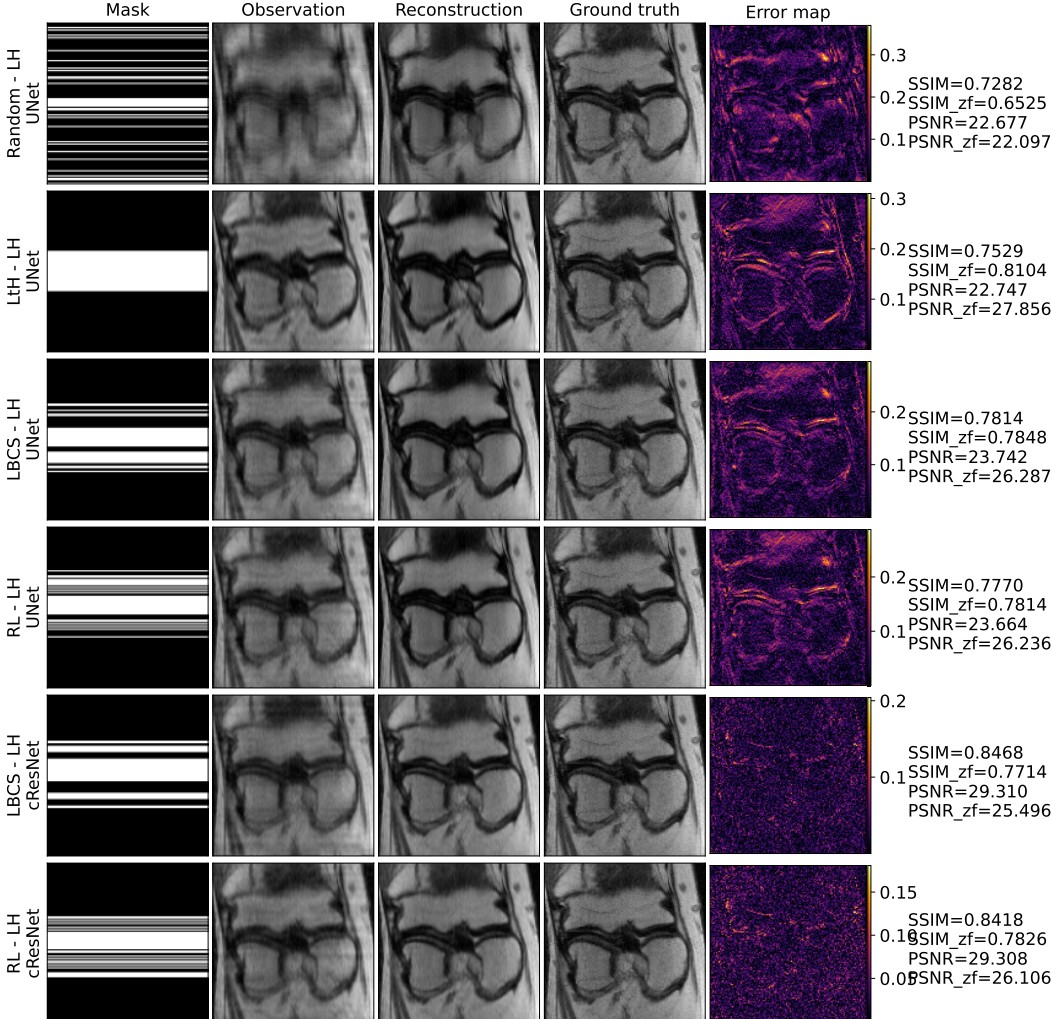

Figure 8: Visualization of masks, observations, reconstructions and ground truths and error maps (|reconstruction − ground truth|) at 25% sampling rate, for different policies (Random, LtH, LBCS, RL). The data are processed according to the c+rhz setting, i.e. cropped then resized images, horizontal undersampling and Zhang-type distribution of masks. The SSIM and PSNR values are given on the right, and here, *zf* refers to zero-filled, and is the SSIM/PSNR taken between the observation and the ground truth.

Focusing first on Figures 8 and 9, we can observe that random and low-to-high sampling lie significantly behind the performance of LBCS and RL. This is the reason why they are not included in the rest of the figures. We can see on the observation that random sampling tends to miss important structures, and results in a severely aliased observation. On the contrary LtH, that focuses on low frequencies, obtains a good quality observation, but fails to yield an improvement after reconstruction, and generally loses out on higher frequency details, yielding a poorer performance especially on edges. In the rest of the comparison, it is hard to notice any significant visual difference between the images obtained by LBCS and the RL method of Bakker et al. (2020). It is however clear that in the

`c+rhz` setting, the cResNet yields a significantly better and sharper reconstruction at 25% sampling, compared to the UNet. This is also confirmed by the results in Table 7. On this particular image, this trend is also observed at 12.5% sampling rate on Figure 9, but this is *not* a consistent trend, as this is not highlighted by the AUC computation of Table 6.

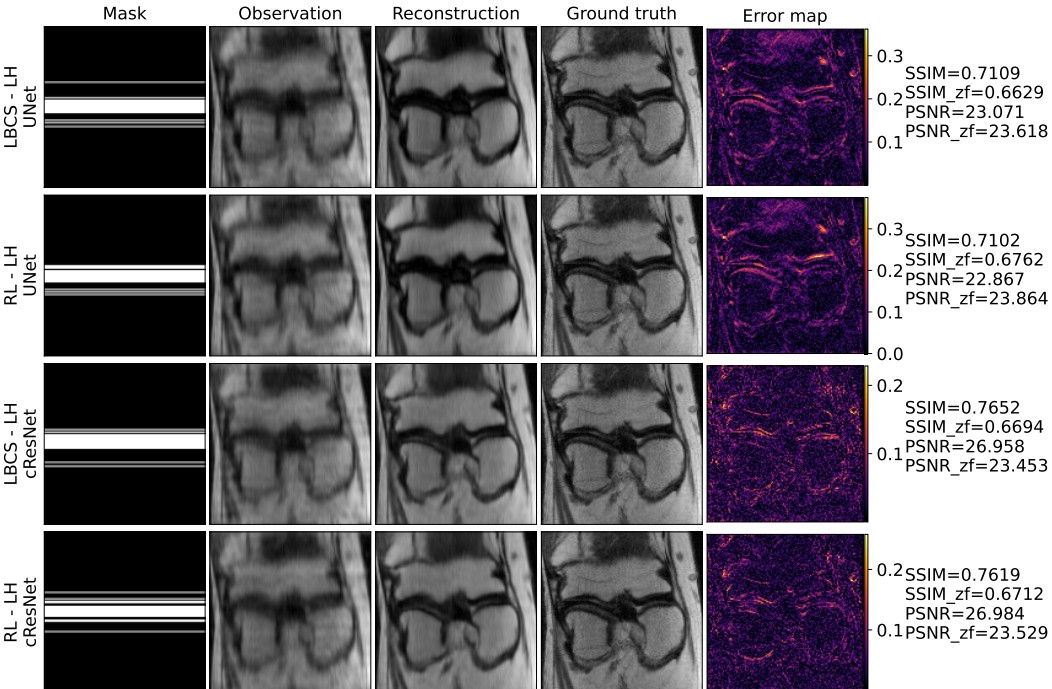

Figure 9: Additional visualization for the image displayed in Figure 8. The results feature 12.5% sampling rate.

Turning now to Figure 10, it is interesting first to discuss the policies obtained by LBCS and RL respectively. The LBCS policy is fixed for the reconstruction algorithm, so the first and third rows of Figures 10a and 10b will each feature the same policy. The adaptive RL policy, on the second and fourth rows seems to have central backbone of common frequency, but varies more around higher frequencies, as observed in the Figure 5 of the appendix of Bakker et al. (2020). However, in both cases, these differences have very little quantitative impact, and the same is true visually: there is no clear visual difference between the reconstructed images at either stage.

While this is not an exhaustive visual investigation, and does not directly assess the suitability of the different policies for various downstream tasks, this lack of visual difference could suggest that RL might not bring a significant improvement over simpler techniques such as LBCS in such cases. However, the question remains open in the case where the sampling policy would be tailored directly for the downstream task, rather than optimized for reconstruction quality, but this falls beyond the scope of the current work.

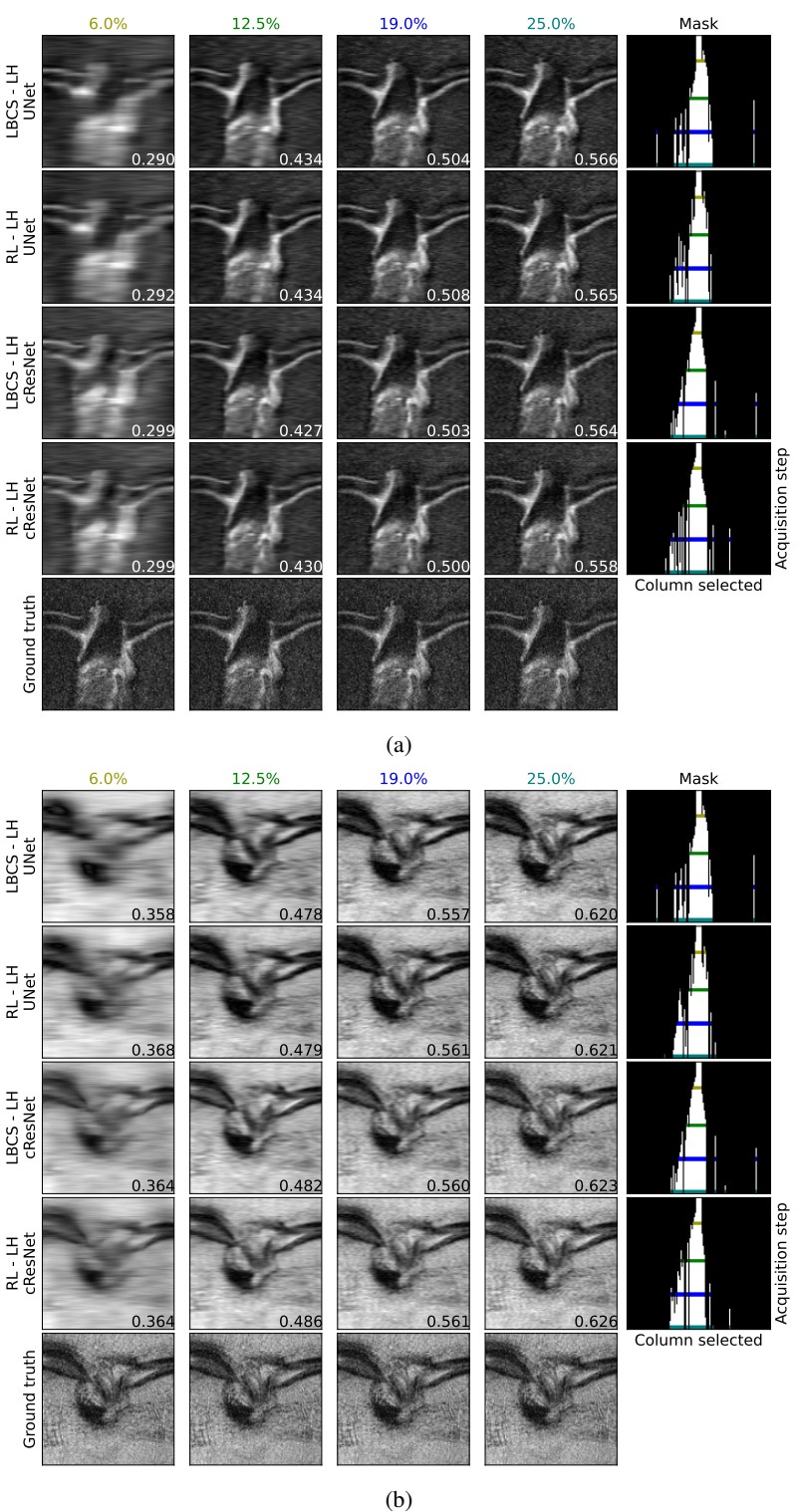

Figure 10: Visualization of reconstructed images at different sampling rates (6%, 12.5%, 19% and 25%) for two sampling policies (LBCS and RL) and two reconstruction algorithms (UNet and cResNet). The data are processed according to the `cvb` setting, i.e. cropped images, vertical undersampling and Bakker-type distribution of masks. The last row shows the ground truth (repeated), and each reconstruction has the corresponding SSIM displayed at the bottom right of the image. The rightmost column display the columns acquired during sampling (in white) as as a function of the acquisition steps: starting on top with only center frequencies and progressively adding more and more lines to the sampling mask. The top plot (a) displays a proton density, fat saturated (PDFS) image, and the bottom plot displays a proton density (PD) image (Zbontar et al., 2019).

# H INITIAL EXPERIMENTAL RESULTS ON PINEDA ET AL. (2020) TRAINED ON THE SETTING OF BAKKER ET AL. (2020)

This appendix contains initial results on training the policy model of Pineda et al. (2020) on the data processing setting of Bakker et al. (2020), in order to provide a clearer comparison between both methods. This experiment was started during the rebuttal period, and could not be carried out entirely, due to the duration of training required for the RL policy model to converge[4]. This comparison is then *not* fully representative of the performance of the model. However, this highlights again an advantage of LBCS over RL methods in its simplicity and quick training.

We train the model on the cropped field of view, with vertical masks. The images are of size $128 \times 128$, and the acquisition ranges from 4 to 32 columns, which corresponds to the long horizon setting. The normalization used throughout training is the image normalization of Zbontar et al. (2019), instead of a constant scaling, as descried in Pineda et al. (2020). Using a smaller field of view enabled to carry out some parameter changes, namely using larger DQN batch size (64 instead of original 2, $\times 32$) with a correspondingly larger replay buffer (100k instead of original 20k, $\times 5$) and run 100 episodes in parallel instead of 2 (in order to refresh the replay buffer in accordance with the larger batch size and size, $\times 50$). With the larger batch size, we allow a larger learning rate of $0.004$ instead of $0.001$ (to account for the batch size and less updates, $\times 4$). We originally planned to use 160k updates (with our batch size of 32x that of the original, which would correspond to the order of magnitude of transitions used in Pineda et al. (2020)), but this did not suffice for the model convergence, and have the model run till 1.6 million transitions. We set the epsilon greedy exploration to decay to the final value within 32k updates (2e6 samples with our batch size).

We observed convergence to an average episode reward in around 775k updates, (roughly 50e6 samples) but it continued to slightly improve. We think that using a longer epsilon decay time and more DQN steps between collection rounds might improve performance in a data efficient way, similar to the observations of van Hasselt et al. (2019).

The SSIM and PSNR AUCs against the sampling rate are shown below. The SSIM-AUC ($0.4342$) and PSNR-AUC ($20.705$) are below that of the random sampling run in Tables 6 and 8 respectively, so it is clear that the training did not succeed.

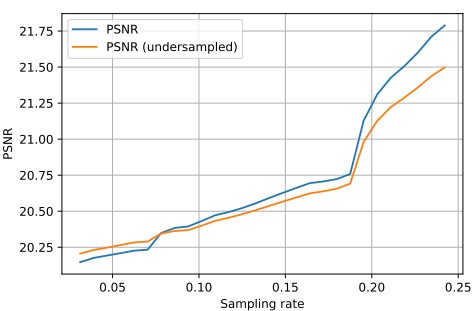
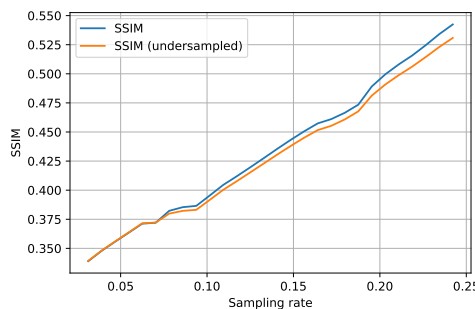

(a) PSNR vs. sampling rate of SS-DDQN.      (b) SSIM vs. sampling rate of SS-DDQN.

Figure 11: Performance plots for the SS-DDQN. This was done in the cropped, vertical and Bakker-type mask setting (`cvb`).

---

[4]The authors noted training times ranging from 7 to 20 days until convergence

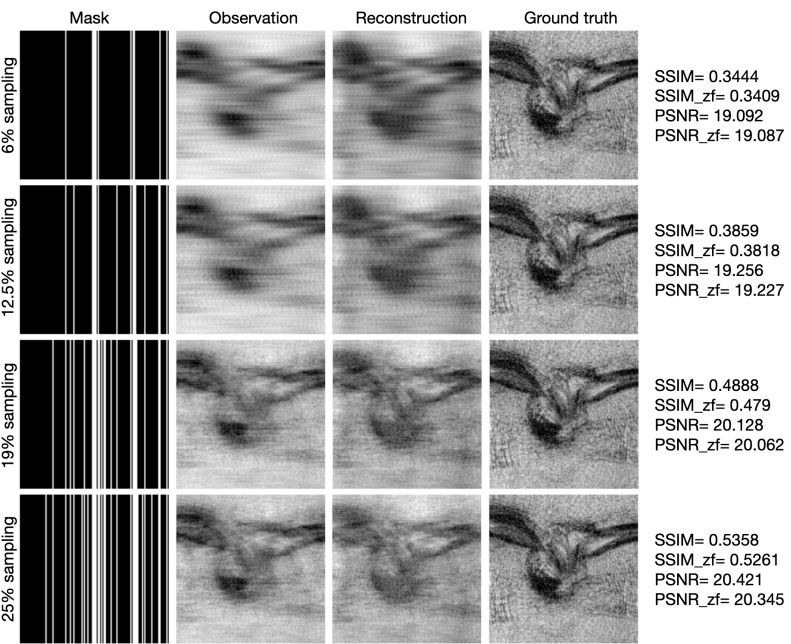

Figure 12: Visual comparison between observation, reconstruction and ground truth of the SS-DDQN in the cropped, vertical, Bakker-like mask setting (`cvb`). The images evaluated are the same as in Figure 10.

