# OpenReview forum: "On the benefits of deep RL in accelerated MRI sampling"
_ICLR.cc/2022/Conference — ICLR 2022 Submitted_

### Official Review · Reviewer_8cse · 2021-10-31

**Correctness:** 3
**Technical Novelty And Significance:** 1
**Empirical Novelty And Significance:** 3
**Recommendation:** 5
**Confidence:** 2

**Main Review:**

Pros:
The paper studies the real world problem of how to accelerate the acquisition of MRI scans and whether RL methods can help in this scenario. The paper considers two relevant RL approaches to the MRI acceleration task and compares it to a strong non-DL baseline (LBCS). The paper presents empirical results for a large variety of settings using different preprocessing and postprocessing options.

Cons:
- The paper sets out to unify the results of recent improvements in MRI acceleration but does not offer comparisons of the two considered RL methods. This generally makes it hard to objectively judge the presented results in the paper. The paper would benefit from an "apples to apples" comparison of the different aspects that are being considered including all steps of preprocessing, sampling strategy, and postprocessing.
- The paper mentions that different previous works rely on different possible metrics and that different metrics lead to different conclusions. The display of the results is not very clear to me, e.g.: What's the difference between the AUC for sampling rate or acceleration rate in Table 3? How are they computed? As such it would be useful to present the results of all experiments with all possible metrics and state which metrics would be favourable to use in which scenario. Metrics such as MSE are mentioned but don't seem to be reported.

Minor:
- Reversals are described as cases where the RL approach is suddenly outperformed by LBCS. However, later it is enough that the performance of the RL methods are matched. This leads to misunderstandings and seems to misrepresent the performance gains.
- The display of the results in the table would benefit from reiteration of the design choice made. Rather than just using "cvb" it would be easier to use "cropped, vertical, Backer parameters or explicitly offer an explanation in the caption. Alternatively, one could add a table to explain the possible options for the different design axes.
- There are various typos in the paper:
  - Just after eq. 3, "estimates from training data" -> "estimated from ..."?
  - Page 6, "Except fir the deterministic LtH and NA Oracle,we report" -> missing space after the comma
  - Page 8, "and of the corresponding AUC consists of 20 lines out of the 100 acquired. While is true..." -> "While it is true"?

**Summary Of The Paper:**

The paper investigates the effectiveness of deep RL methods for accelerating the acquisition of MRI scans. It compares two recent methods by Bakker et al. as well as Pineda et al. to non-RL baselines. The results indicate that deep RL provides little benefits over the LBCS baseline.

**Summary Of The Review:**

My main issue with the paper is that it is hard to follow the analysis. It would benefit the paper to combine the analysis of both RL methods and rather ask questions about which design choices have the highest impact. Similarly, the analyses should incorporate the different choices of evaluation metrics. This would enable the authors to propose a recipe of how to approach the problem of MRI acceleration.

---

> ### Author Response · Authors · 2021-11-10
> **We thank the reviewer for the comments and helpful suggestions, but would like to provide some clarifications**
>
> We thank the reviewer for their review, pointing out typos and suggestions for improvement.
>
> We mainly agree that there is benefit of an apples-to-apples comparison between all three methods. Would you agree that training the DQN in the setting of Bakker et al  (cvb) is sufficient to establish this comparison?
>
> However, we need to stress that the point of this paper isn’t to _fix_ RL for MRI sampling but to point out that previous publications failed to include some surprisingly strong baselines and so should be recontextualised (see also our joined comments). As such “proposing a recipe of how to approach the problem of MRI acceleration” is slightly out of scope of our paper, since we did not set out to develop a new sampling method, simply quantify the added value (or lack thereof) of recent SotA methods .
>
> **AUC explanation.**
> We will also add an appendix about the computation of the Area Under Curve (AUC). The AUC curves of Table 3 were computed from Figures 3 and 6, respectively 3a and 6a for the sampling rate and 3b and 6b for the acceleration factor. The AUC is computed on a given interval (the x-axis in the figures), which implies that moving from the sampling rate to the acceleration factor requires remapping the points on the graph, and changing the area under the curve by extending greatly the contribution of high acceleration factors (low sampling rates): 20 sampling locations cover 90% of the acceleration factor plot. As the AUC summarizes the plot, it will be greatly affected by the emphasis put on high acceleration factors.
>
> **Regarding the reviewers remaining concerns.**
> - Since the point of the paper is _not_ to compare which RL method benefits from which hyperparameter and preprocessing variation and how to get the most out of RL but rather that the benefits of RL over a fixed mask are so marginal that they seem to lie under the noise floor, we think that the ablations between Bakker et al. and our fixed method are sufficient. If the reviewer disagrees we are open for arguments and are happy to discuss/add relevant ablations
> - We will incorporate the comments about reversals and the display of the results

---

> > ### Comment · Reviewer_8cse · 2021-11-18
> > **Thanks for clarification.**
> >
> > Thanks for the clarifications - I think I now understand the differences in the AUC calculation. I agree that this paper does not need to "fix" the shortcomings of previous works.
> >
> > As for my other points. I agree that this paper does not need to solve all of MRI sampling nor fix the shortcomings of prior work. However, the contributions of this rely on an empirical analysis to show that LBCS performs at least as well as SoTA RL methods. As such I believe that the paper shouldn't shy away from providing extensive empirical analyses of various settings. So what is required to cover necessary comparisons?
> >
> > In my opinion, an ideal comparison would show the outer product of compared settings: {Bakker, Pineda, LBCS} x preprocessing x {U-Net, cResNet} x metrics. I am aware that this might be prohibitive in terms of computational cost and one might need to slim this into an apples-to-apples comparison with added details for the parts that are two expensive to run in every setting. Adding the DQN in the cvb setting is certainly a good start.
> >
> > That leaves the question about metrics. In the paper, it is stated that different metrics highlight different aspects of a methods performance. Different metrics are mentioned but only a small subset are actually computed. It would be interesting to calculate the full set of metrics for all methods reported in the paper as one might see different behaviours in the different metrics. Are there general statements one can make about the benefits or drawbacks of a method that certain metrics pick up on, e.g. (just for clarification, those statements aren't true) "methods with better MSE capture high-frequency information better, whereas methods with better SSIM capture low-frequency information better" and hint at which metrics to optimise for in different application scenarios.

---

> > > ### Author Response · Authors · 2021-11-19
> > > **Thank you for the reply**
> > >
> > > Thank you for the reply - As mentioned in the revision thread, we have provided a clarification of the AUC computation in Appendix A.4
> > >
> > > Regarding the empirical evaluation, we agree that empirical analysis is important, which is why we strove to include as detailed experiments as possible. Indeed, we originally set out to check the outer product you described.  However, despite having access to a DGX100, the estimated training time for the DDQN is **7-20 days**, in line with the running time of the original work which we confirmed in personal communication with the authors to rule out misconfiguration on our side. The fact that LBCS can be scaled to run on the same machine to yield results within minutes or a few hours further illustrates the questionable ROI of deep RL for this task.
> > >
> > > Due to this prohibitive cost, we initially chose to not include Pineda in the ablations and we would argue that while desirable, for most academics such an ablation is impossible or at least has a bad ROI due to the prohibitive computational expense and the diminishing return after verifying one or two settings. We have been running the training for the cvb-DDQN since our initial response and will be reporting the performance but will need to stress that the performance might not be fully representative due to
> > >
> > > 1. not having the time to tune the hyperparameters appropriately
> > > 2. the above mentioned training time since the authors mentioned that even after 20 days of training there can continue to be slight improvements
> > >
> > > Regarding the metrics, our point is less about the underlying metric than the summary statistics of a sampling curve (AUC vs. average vs final rate). We have clarified this point in the paper in our latest revision and have added a PSNR evaluation (Appendix C.3) of the models that we run with Bakker’s RL policy in the appendix, for completeness. As PSNR, MSE and NMSE are conceptually close (PSNR is roughly -log of the MSE), we observed that the predictions of these three metrics are generally aligned, which also seems to be the case in the work of Pineda et al. [1].
> > >
> > > Generally, metrics such as MSE, NMSE, PSNR and SSIM are used due to their ease of computation, and give limited insight into diagnostic quality [3].  MSE, NMSE and PSNR only rely on pixel differences, evaluating pixels independently, while SSIM relies on local patches, including more of the image structure in the metric [2]. It has been observed, on the fastMRI 2019 challenge, that SSIM showed a similar trend to the radiologists’ rating, while other metrics “showed almost opposite trends” [4]. However, all metrics tend to promote smooth images [4] but it is widely known that MSE, NMSE and PSNR promote low-frequencies/high-energy components, which was also noted by Bakker et al. [5]. However, finding a good perceptual metric that is also easily differentiable is a challenging research topic, with some work done for general images [6] but in our experiments LPIPS did not yield sensible results for the type of distortions induced by sampling. Does this answer your question?
> > >
> > >
> > > [1] Pineda, Luis, et al. "Active MR k-space sampling with reinforcement learning." International Conference on Medical Image Computing and Computer-Assisted Intervention. Springer, Cham, 2020.
> > > [2] Zbontar, Jure, et al. "fastMRI: An open dataset and benchmarks for accelerated MRI." arXiv preprint arXiv:1811.08839 (2018).
> > > [3] Knoll, Florian, et al. "Advancing machine learning for MR image reconstruction with an open competition: Overview of the 2019 fastMRI challenge." Magnetic resonance in medicine 84.6 (2020): 3054-3070.
> > > [4] Muckley, Matthew J., et al. "State-of-the-art Machine Learning MRI reconstruction in 2020: Results of the second fastMRI challenge." arXiv preprint arXiv:2012.06318 (2020).
> > > [5] Bakker, Tim, Herke van Hoof, and Max Welling. "Experimental design for MRI by greedy policy search." Advances in Neural Information Processing Systems 33 (2020).
> > > [6] Zhang, R., Isola, P., Efros, A.A., Shechtman, E., & Wang, O. (2018). The Unreasonable Effectiveness of Deep Features as a Perceptual Metric. 2018 IEEE/CVF Conference on Computer Vision and Pattern Recognition, 586-595.

---

> > > ### Author Response · Authors · 2021-11-23
> > > **Training did not succeed, we see this as a case in point of the high complexity of RL and can add a run the original hyperparameters to the camera ready.**
> > >
> > > Dear reviewer,
> > >
> > > We have added Appendix H with a comparison to our best effort reproduction to Pineda et al. in the setting of Bakker et al., as suggested by reviewer 8cse. The SSIM and PSNR AUCs against the sampling rate are shown in the table below.
> > >
> > > As we have noted, the computational and time expense by this method made reproduction from scratch difficult (the authors noted training time 7-20 days in communication with us). We therefore have to stress (as we do in the paper) that this cannot be a fair comparison with the final performance of the method, however we also want to highlight this as a point in the favour of our paper: we have access to above average compute and are experienced in working with RL and ML methods, nonetheless we were not able to transfer the method to this new setting in the limited time due to the time required for debugging, training and tuning this method.
> > >
> > > While we are confident we can tune the method to work well in our setting, this is an example of the fact that RL methods are notorious for requiring careful tuning and even small and reasonable changes in parameters can make the results unusable. Meanwhile the LBCS baseline we studied is much simpler and in our experience very robust to hyperparameters as long as a simple rule of thumb of using about 25% of the sampling candidates and a batch size on roughly the order of magnitude of the dimensionality is used (for best performance, it remains strong even with cheaper settings).
> > >
> > > The changes we performed consist of
> > >
> > > - The images are 128x128 and we sample from 4 to 32 columns, meaning our action space is about a quarter of that of Pineda et al. We normalize consistent with the cvb setting.
> > > - The smaller image size allows a larger DQN batch size (64 instead of original 2, 32x)...
> > > - ….which we accommodate with a larger replay buffer (100k instead of original 20k, 5x)....
> > > - ...and by setting num_parallel_images 100 instead of 2 (in order to refresh the replay buffer in accordance with the larger batch size and size, 50x)
> > > - With the larger batch size, we allow a larger learning rate of 0.004 instead of 0.001 (to account for the batch size and less updates, 4x)
> > > originally we planned to use 160k updates (with our batch size of 32x that of the original, this would train on roughly the same number of real transitions as pineda et al), however since we were nowhere near convergence we have let the model run till 800k(1.6e6 tomorrow) and are continuing the training
> > > - we set the epsilon greedy exploration to decay to the final value within 32k updates (2e6 samples with our batchsize)
> > >
> > > We observed convergence to an average episode reward in around 775k updates,  (roughly 50e6 samples) but it continued to slightly improve. We report the final performance at 1.e6 transitions (102e6 samples). We think that using a longer epsilon decay time and/or more DQN steps between collection rounds might improve performance in a data efficient way, similar to the observations of [1] but did not have time to test these assumptions due to the constraints given by the rebuttal.
> > >
> > > Again, we stress that we do not claim that this is a fair comparison. Given that it would be impossible to replicate the authors hyperparameters in the rebuttal time (which is less than the 20 days required for full convergence) even if we had started at day 0, we propose that we launch the original hyperparameter training and include this ablation in the final version of this paper as a condition of acceptance.
> > >
> > > For production, the question of ROI hinges on the question whether DDQN is likely to achieve a stellar margin if tuned properly. However as we showed in the main paper already, even when comparing against it tuned by the authors and in its own setting, the baseline can match or outperform it. Because of this, we find it unlikely to expect a dramatic return on investment and that the computational expense would not be worth it even if one can afford the investment.
> > >
> > >
> > > This means that had this surprisingly strong baseline been included, the paper most likely wouldn’t have been published without additional changes and that our paper has value for the ML community to correct the record on the SotA of methods studied at ICLR and other ML conference.
> > >
> > > | SSIM AUC | PSNR AUC |
> > > | --------------  | ---------------  |
> > > |  0.4342     | 20.705 |
> > >
> > > [1] van Hasselt, Hado P., Matteo Hessel, and John Aslanides. "When to use parametric models in reinforcement learning?." Advances in Neural Information Processing Systems 32 (2019): 14322-14333.

---

### Official Review · Reviewer_zYDG · 2021-11-02

**Correctness:** 3
**Technical Novelty And Significance:** 2
**Empirical Novelty And Significance:** 2
**Recommendation:** 3
**Confidence:** 4

**Main Review:**

Pros:
- The paper focuses on a clinically relevant topic
- The paper presents a solid comparison
- The writing is clear and easy to follow.

Cons:
- There is a lack of methological novelty.
- There is a lack of visual comparison. Although visualizations are somehow subjective, the reconstructed images are important for doctors to compare lesions/patterns in medical diagnosis and hence visual comparions are important. Will these different Fast MRI Recon methods result in similar visual quality (lesion details/organ boundary shapes) or they show different reconstruction performance for different tasks (lesion detection/organ checking)? All these answers should be asked by comparative visual readings.

Typos:
"following equation equation 1"  in the last line of  Paragraph 5, Section 3;
“preprocessing” ->  “post precessing” in the last line of Paragraph 7, Section 3.


**Summary Of The Paper:**

The paper first surveys several recent Fast MRI Recon methods, including RL-based and greedy-policy-based ones, that reconstruct high quality images from highly undersampled data from the perspective of optimizing the sampling of k-space data and then concerns the RL-based methods. Further it considers a typical Fast MRI Recon framework that contains three main blocks of preprecessing (cropping and/or resizing, using magnitude only etc.), subsampling (random, LtH, LBCS, adaptive, etc.), and post-precessing and reconstruction, and evaluates its performance changes triggered by seemingly trivial changes at even a single of these stages in a series of experiments. It shows that the RL-based method, which adpatively computes the sampling mask, does not offer benefits over fixed masks.

**Summary Of The Review:**

Overall, the paper contains certain empirical value, arguing that RL-based approach is not effective. But it lacks the typical methological novelty necessary for an ICLR paper.

This paper is more suited for MICCAI or medical imaging related conferences.

---

> ### Author Response · Authors · 2021-11-10
> **We thank the reviewer for their comments and valid suggestion, however could you please be more clear in your criticisms?**
>
> We thank the reviewer for their review and pointing out the typos.
>
> However, it is unclear what they mean with a “lack of methodological novelty”?  We show that the current publication standard for “methodological novelty” are insufficient since 2 papers which were considered to bring a novel methodology for improved performance *did not* in fact improve over a very simple (if surprisingly strong) baseline. We think that this is an important result for the ICLR community in and off itself, see also [our joint comment](https://openreview.net/forum?id=fRb9LBWUo56&noteId=xSuAbr2KXoq). If the reviewer disagrees we would like to ask them to be more detailed in their reasoning.
>
> We agree with the point about the visual comparison and will provide one in the rebuttal period. Would you be satisfied with a comparison of the same image at different points of the sampling process?

---

### Official Review · Reviewer_PeJ1 · 2021-11-04

**Correctness:** 4
**Technical Novelty And Significance:** 2
**Empirical Novelty And Significance:** 2
**Recommendation:** 3
**Confidence:** 2

**Main Review:**

Strengths:
The authors are commended for recreating results and systematically evaluating previous works on the topic.
The authors were charitable in their presentation of Bakker and Pineda.
The Conclusion section is very good. They provide a clear and distinct summary of their contribution in their work AND charitably restate the limitations of the discussed works. I think this was a graceful presentation of the limitations of other investigators’ works without being overly critical.
The authors did a great job pointing out the importance of openness and reproducibility. This is absolutely critical to good research and reproducing results. The authors are commended for pointing this out in the paper explicitly!

Weakness:
While I think the authors report on an important topic, the work lacks a strong enough negative or positive contribution to meet the threshold for publication in ICLR. The authors correctly demonstrate the limitations of deep RL in accelerated MR sampling; however, the previous authors also stated these limitations. The ROI on any clinically relevant ML system can be called into question due to computational costs. However, these costs are in no way insurmountable and rather simple to overcome in practice.

The lack of performance boost is relevant and the authors point this out. However, this fact alone is insufficient to warrant publication here and is better suited for another venue.


**Summary Of The Paper:**

The authors report on the use and current limitations of using deep reinforcement learning for accelerated sampling in MR imaging. They focus on two specific works that use deep RL for sampling: Pineda and Bakker. They mainly compare these methods against a previously described non-RL technique: Stochastic Learning-based Compressive Sampling (LBCS). This method trains a non-adaptive, greedy sampling policy that selects as a measurement candidate in each acquisition step the column that leads to the greatest average improvement over a sample from a training dataset.

The authors conclude that the ‘return on investment’ of adopting RL over LBCS is generally marginal compared to what changes in the modeling pipeline can make. Improving the reconstruction architecture or using masking regimes adapted to the sampling horizon yield more significant gains than using RL-based methods. The authors state that their results confirm the stated limitation in both Pineda and Bakker, namely that neither adaptivity nor long-term planning methods are currently worth the increased cost.


**Summary Of The Review:**

Overall, I am in complete agreement with the motivation and sentiment of the paper, but as presented, it does not make a sufficient contribution to warrant publication.

---

> ### Author Response · Authors · 2021-11-10
> **We thank the reviewer for their kind words but would like to ask for clarification what would make the paper a "sufficient contribution"?**
>
> Thank you for your review, and the strengths that you highlighted in our paper (“...commended for recreating results and systematically evaluating previous works on the topic...”, “...are commended…”,”...report on an important topic…”) . However we are confused how these sentiments translate into your rating, could we ask you to elaborate what could be added to the paper to make it “a sufficient contribution to warrant publication” ?
>
> As discussed also in the general response, we believe that our work contains a strong negative result, as if either of the works of Pineda and Bakker had included the LBCS baselines, their papers might not have been accepted, due to only marginal  performance gains. This result was quite surprising to us, and testifies to the unexpected strength of the baseline. Moreover, the reversals that occur when changing the data processing pipeline also raise a methodological red flag to the community, calling for more extensive evaluations and investigations when considering the problems of reconstruction and sampling in accelerated MRI.
>
> In addition, we also want to stress that our work goes beyond the limitations highlighted by the previous authors: while Pineda showed that adaptivity was not always necessary, they still used long-term planning. Similarly, Bakker used greedy training and insisted on the importance of adaptivity. If any of the previous authors had stated the limitations that neither long-term planning nor adaptivity seemed to matter, it is likely that they would not have published their works.
>
> We agree with the reviewer that the ROI on medical imaging due to computational costs is easy to call out, and we want to highlight that our contribution goes beyond this: the issue does not mainly lie in the fact that RL require additional computation, but that current results cannot clearly highlight that they bring a consistent improvement over simpler greedy methods if one invests these resources. Stated differently, we are not attacking the methods for being expensive, we are pointing out that for all the expense they require, there are not any decisive gains.

---

### Official Review · Reviewer_vo6W · 2021-11-07

**Correctness:** 3
**Technical Novelty And Significance:** 1
**Empirical Novelty And Significance:** 2
**Recommendation:** 5
**Confidence:** 3

**Main Review:**

strength:
1. This paper works on a very important topic, accelerating MRI by optimizing signal sampling.
2. This paper compares several SOTA RL methods to present an interesting conclusion.

weakness:
1. I'm not convinced by the novelty of this paper. Honestly, this paper may be more suitable in a medical imaging journal.
2. I'm not sure if this research is really meaningful. Does accelerating MRI really need RL for signal sampling? I am not optimistic that RL based sampling methods can be really applied in MR imaging industry.

concern:
1. As for the experiments, are these experimental settings well set for a fair comparison?



**Summary Of The Paper:**

This paper explores the benefits of current deep Reinforced learning in accelerating MRI sampling and concludes that the benefit brought from current SOTA RL methods is not aligned with the computational complexity of these methods.

**Summary Of The Review:**

Though this paper donot introduce quite new algorithm, it is still valuable since it concludes the recent RL based sampling methods in accelerating MRI and analyze the benefit. However, I'm not sure if this paper is suitable for submission in ICLR. Maybe it is more suitable in a medical imaging journal.

---

> ### Author Response · Authors · 2021-11-10
> **Thank you for the postitive comments, could you please be more specific in your criticisms?**
>
> We thank the reviewer for their comments. We are a bit confused by this review as it expresses a positive sentiment ( “...it is still valuable”) and makes no criticisms justifying the  “not suitable for ICLR” score. As highlighted in the common reply, we believe that we show convincing negative results about the practical performance of current RL methods in accelerated MRI.
>
> We believe there is a misunderstanding. We wholeheartedly agree with the reviewer that it seems that RL is not the way to go for MRI sampling, which is in fact the main statement of our paper: RL might not be required in MRI because it does not bring performance gains for increased complexity. Nonetheless, papers about this have been published in top ML conferences and this is a very active area of research (see the references [1-3] below for a few recent works in ML venues in addition to Bakker and Pineda), which is why we think it is important to be published at a venue like ICLR so future publications with weak baselines can be avoided [(see our general comment)](https://openreview.net/forum?id=fRb9LBWUo56&noteId=xSuAbr2KXoq). If the reviewer does not agree with us, can we ask for a further justification why?
>
> Regarding the doubts on whether our experiments are suitable for a fair comparison, we put a lot of effort into ruling out any confounders (we also refer to the ablations in the appendix). Does the reviewer have any specific concerns? We are happy to address them in the rebuttal period.
>
> - [1] Alkan, C., Mardani, M., Vasanawala, S., & Pauly, J. M. (2020, December). Learning to Sample MRI via Variational Information Maximization. In NeurIPS 2020 Workshop on Deep Learning and Inverse Problems.
> - [2] Yin, T., Wu, Z., Sun, H., Dalca, A. V., Yue, Y., & Bouman, K. L. (2021, May). End-to-End Sequential Sampling and Reconstruction for MR Imaging. arXiv preprint arXiv:2105.06460.
> - [3] Van Gorp, H., Huijben, I., Veeling, B. S., Pezzotti, N., & Van Sloun, R. J. (2021, July). Active Deep Probabilistic Subsampling. In International Conference on Machine Learning (pp. 10509-10518). PMLR.

---

### Author Response · Authors · 2021-11-10
**Joint reply (part 1 of 2)**

We thank the reviewers for the overall positive spirit of their constructive comments, if not their scores. All reviewers acknowledged that the results we obtained are important for the assessment of ML methods in accelerated MRI. There are some requests for clarifications and additional experiments which we address in individual comments.

What remains are general sentiments that the results are “not enough” and that ICLR might not be a suitable venue. We would like to address this or at least ask all reviewers to provide more details on their reasoning and proposals on what would make the results sufficient for acceptance?

In the following, we argue our case on why we think our paper is suitable for acceptance at ICLR based on the following quote from the guidelines of ICLR: https://iclr.cc/Conferences/2022/ReviewerGuide


>Submissions bring value to the ICLR community when they convincingly demonstrate new, relevant, impactful, or insightful knowledge. Submissions can achieve this without achieving state-of-the-art results.

Our work shows that methods which were previously considered state-of-the-art by leveraging long range adaptive sampling policies do not in fact bring any improvement over a fixed, greedily trained sampling mask constitutes new, relevant, impactful and insightful knowledge. This hints towards either further work to be done for RL methods to unlock their full potential or that the assumptions common in our community (being from the ML and CS community, not the MRI community ourselves) that adaptivity and long range planning should always bring improvements do not hold in this case. To us, this is quite surprising, which is why we prepared our work to focus only on this issue.

We also observe that medical imaging, in particular MRI has been a prime application in top ML conferences, and many of the related works have been published at ICLR (e.g. [1]) and other top ML conferences (Bakker et al. [2]  and others [3] at NeurIPS, Darestani et al. at ICML [4]). In line with such works, it seems relevant to aim at publishing our findings in an ML community, as we think that the points that we raise can primarily benefit ML practitioners looking to apply their skills to accelerated MRI.

We were also motivated to submit our work to ICLR as the conference has previously published works, such as [5], that focus on analyzing existing methods without proposing technical methods and simply improves the communities understanding of how to judge and build on the existing works. [5] also features no novel algorithm and has its contribution squarely in establishing the differential value of each component they studied. The main difference between [5] and us is we find that the improvements were entirely virtual (i.e., dependent on the reconstruction method and the specific setting, not the sampling method)  when comparing with a strong baseline using *the same* reconstruction method in the *same* setting, but our conclusion is the same: a call to action for more careful evaluation as well as better understanding of the factors that determine performance.

We also think that in order to alleviate publication bias it is important the community highlights papers that recontextualize prior work and shows their improvements over available baselines are marginal if performed properly just as much as it highlights the initial works. The computer vision ML community recently did this with metric learning [6] and since ICLR and similar conferences like NeurIPS and ICML highlight MRI sampling methods we think our paper fills a similar niche.

We therefore think that our contribution important and relevant for the ICLR and broader ML community 1) as a piece of the scientific process in order to avoid the publication bias towards publishing positive results and also 2) in order to highlight pitfalls for ML researchers interested in developing methods in this domain that could lead their comparisons to be similarly subtly defective as the ones we studied in this work. For example we highlight some minute issues with the preprocessing of Bakker et al. which did not occur in the case of Pineda et al., which suggests that there is still a need to raise awareness about these subtle difficulties.



[1] Huijben et al.. Deep probabilistic subsampling for task-adaptive compressed sensing, ICLR 2020
[2] Bakker et al. Experimental design for MRI by greedy policy search. NeurIPS 2020
[3] Defazio et al. MRI Banding Removal via Adversarial Training. NeurIPS 2020
[4] Darestani et al.  Measuring Robustness in Deep Learning Based Compressive Sensing. ICML 2021
[5] Engstrom, Ilyas et al. Implementation matters in deep policy gradients: A case study on ppo and trpo, ICLR 2019.
[6] K. Musgrave, S. Belongie, and S.-N. Lim, “A Metric Learning Reality Check,” in Computer Vision – ECCV 2020, Cham, 2020, pp. 681–699. doi: 10.1007/978-3-030-58595-2_41.

---

> ### Author Response · Authors · 2021-11-10
> **Joint reply (part 2 of 2)**
>
>
> In addition, we argue that our work has general applicability beyond MRI sampling since it highlights the need for understanding under which conditions and how adaptivity, long term planning and training the sampling method on the used reconstruction method will improve performance (e.g., what exactly is it about Fourier space that apparently renders adaptivity and long term planning only marginally useful with current techniques? Which domains admit smarter sampling methods that have a higher return on investment?), which is a general question for the ML community of which this paper presents a case study.
>
>
> Finally, while we do not want to rule out the possibility that the methods we studied could be made to outperform our baseline, we believe that if the works had been submitted with this additional baseline, they might not have been accepted, as reviewers would certainly have pointed out that these RL methods add complexity without significantly improving on the baseline. We believe that this result does not testify to a lack of diligence by the original authors, but rather to a surprisingly strong performance of LBCS in accelerated MRI.
>
> To this end, we ask the reviewers the following question: do you think that an ML researcher will have a correct model of the world in their head if they only read the works of Pineda et al. and Bakker et al. showing the benefits of RL for sampling, but not our work?
>
> - [1] Huijben et al.. Deep probabilistic subsampling for task-adaptive compressed sensing, ICLR 2020 [2]
>  - Bakker et al. Experimental design for MRI by greedy policy search. NeurIPS 2020
> - [3] Defazio et al. MRI Banding Removal via Adversarial Training. NeurIPS 2020
> - [4] Darestani et al. Measuring Robustness in Deep Learning Based Compressive Sensing. ICML 2021
> - [5] Engstrom, Ilyas et al. Implementation matters in deep policy gradients: A case study on ppo and trpo, ICLR 2019.
> - [6] K. Musgrave, S. Belongie, and S.-N. Lim, “A Metric Learning Reality Check,” in Computer Vision – ECCV 2020, Cham, 2020, pp. 681–699. doi: 10.1007/978-3-030-58595-2_41.

---

### Author Response · Authors · 2021-11-19
**First revision**

We thank the reviewers once more for their comments. Here are the changes that we made.

**Visual comparison [zYDG].** We carried out analysis on some selected images in App. G, where we illustrate (1.) that there is significant visual difference between basic policies (Random, LtH) and learned ones, but (2.) that the visual difference between LBCS and RL is quite minor, and (3.) that architectural changes lead to much more significant visual changes than a change in policy.  Since the scope of our paper is the unexpected underperformance of a complex ML method (deep RL) w.r.t. a simple baseline and we are ML researchers and not medical experts we do not attempt to judge the medical relevance of the differences.

**Clarifying the term reversals [8cse].**
We clarified our definition of reversals in the introduction of the paper as cases where performance or conclusions are reversed, when RL is matched or outperformed by LBCS. We moved the results from Table 6 about Long Horizon (cvb and c+rhz) to the main body of the paper to illustrate a clear instance of reversal and put them in context with the ones presented in Tables 2 and 3, to illustrate a reversal in conclusions by merely reporting the area under curve instead of the performance at the last step.

We also highlighted the significance of this in the introduction, as reversals allow “to support one’s desired conclusion: that RL outperforms LBCS, that LBCS outperforms RL or that their difference is not significant.” We believe that while researchers would not pick their experiments to suit their desired conclusions, such an issue could arise inadvertently, due to a lack of standards of processing and evaluation in the community.

**Display of results [8cse].** We have clarified the design choices in Tables 2,3 (prev. Table 2) and restated explicitly the experimental settings in Tables 6,7,8,9. We have added the results at the final sampling rate (with standard deviations) as Table 2 and have also added PSNR evaluations in Tables 8, 9.

**Corrected spelling [zYDG, 8cse].** We have corrected the spelling mistakes, thank you very much for the help.

**Definition of the AUC [8cse].**  We have added a definition of AUC in App. A.4.

**Discussion of metrics [8cse]** We have added language to the paper to clarify the difference between underlying metrics (PSNR, SSIM etc.) and summarizing or reporting metrics (AUC, average, final rate), why we focus on PSNR and discuss the subject of metrics and perception in the context of MRI.

**Apples-to-apples comparison with Pineda et al [8cse].** As mentioned in the individual reply to reviewer 8cse, we will add the comparison with Pineda-cvb but the prohibitive computational cost (7 to 20 days of training despite above average compute for the academic context) limits this to a best effort comparison, which strengthens our point of the ROI of deep RL on this task being doubtful. This will be included in our second round of revisions.

**Re-emphasizing our contribution [PeJ1,vo6W].** The revision has also enabled us to clarify the fact that Pineda et al. and Bakker et al. did NOT state the limitations of their respective works but only state *part* of the limitations, leading to mutually exclusive conclusions (even if phrased carefully by both works).

As discussed in the introduction, Bakker et al. found that long range planning does not add much (if any) value but expresses optimism about the benefits of adaptive methods. Pineda et al. finds that  their non-adaptive long range planning method is competitive with the adaptive long range planning method and adaptivity does not add much (if any) value, in direct conflict with Bakker et al. Both papers express optimism about their respective methods and motivate further research.

In contrast, our paper finds that neither adaptivity nor long range planning adds much value over a fixed, greedy policy, casting doubt on the value of further empirical research into deep RL methods on this task and resolving the apparent conflict in the previous work.

This also addresses the concern about the paper belonging to a medical venue: the paper is not about a novel application, but it identifies a task for which RL methods unexpectedly do not give any benefit and opens up the question on why: is there some form of submodularity [1]  at play as the greedy method exhibits near-optimal performance? Is it the special structure in the Fourier space that enables this? What other problems might exhibit such structures and as such be bad candidates for naive application of RL?

We have added language to the conclusion that clarifies this important takeaway.

**Emphasizing practical recommendations [PeJ1, vo6W].** We clearly highlighted the main practical implications of our paper in the conclusion, as four recommendations for practictioners, and discussed these messages in Appendix E.

[1] Krause, A., and Golovin, D. "Submodular function maximization." Tractability 3 (2014): 71-104.

---

### Decision · Program_Chairs · 2022-01-20

**Decision:**

Reject

**Comment:**

While all reviewers acknowledge the relevance of such an evaluation work for the MRI reconstruction field, they all agree that the contribution has a limited fit with a ML conference like ICLR. The work is solid experimentally and will surely interest the audience of conferences like ISMRM or MICCAI. For this reason, the work can unfortunately not be endorsed for publication.